# Barrier properties of Nup98 FG phases ruled by FG motif identity and inter-FG spacer length

Sheung Chun Ng [1], Abin Biswas [2,3], Trevor Huyton[1], Jürgen Schünemann[1], Simone Reber [2] & Dirk Görlich [1] ✉

Nup98 FG repeat domains comprise hydrophobic FG motifs linked through uncharged spacers. FG motifs capture nuclear transport receptors (NTRs) during nuclear pore complex (NPC) passage, confer inter-repeat cohesion, and condense the domains into a selective phase with NPC-typical barrier properties. We show that shortening inter-FG spacers enhances cohesion, increases phase density, and tightens such barrier - all consistent with a sieve-like phase. Phase separation tolerates mutating the Nup98-typical GLFG motifs, provided domain-hydrophobicity remains preserved. NTR-entry, however, is sensitive to (certain) deviations from canonical FG motifs, suggesting co-evolutionary adaptation. Unexpectedly, we observed that arginines promote FG-phase-entry apparently also by hydrophobic interactions/ hydrogen-bonding and not just through cation·π interactions. Although incompatible with NTR·cargo complexes, a YG phase displays remarkable transport selectivity, particularly for engineered GFP[NTR]-variants. GLFG to FSFG mutations make the FG phase hypercohesive, precluding NTR-entry. Extending spacers relaxes this hyper-cohesion. Thus, antagonism between cohesion and NTR·FG interactions is key to transport selectivity.

The nuclear envelope (NE) separates the nucleus from the cytoplasm, confining the exchange of macromolecules to NE-embedded nuclear pore complexes (NPCs). NPCs are elaborate ~120 MDa structures assembled in 8-fold rotational symmetry from 30 different nucleoporins, or Nups for short.

By combining cryo-electron microscopy, X-ray crystallography, and biochemical and genetic approaches, the architecture of the NPC scaffold has been elucidated in remarkable detail[1–7]. The NPC scaffold encloses a ~60 nm wide central channel[6,8,9], which is controlled by a sieve-like permeability barrier to suppress an uncontrolled intermixing of nuclear and cytoplasmic contents[10,11]. Traditionally, two modes of NPC passage have been distinguished, 'passive' and 'facilitated'. The passive passage is efficient for small molecules but already severely restricted for 5 nm (30 kDa)-sized objects. Nuclear transport receptors

(NTRs) are not bound to this size limit. Instead, they can engage in facilitated translocation and carry even very large cargoes through NPCs.

NTRs of the importin β superfamily[12] circulate between the two compartments, draw energy from the RanGTPase system, and pump cargoes against concentration gradients. Importins, such as importin β itself or transportin, capture cargoes in the cytoplasm, traverse NPCs, and release cargoes upon RanGTP-binding in the nucleus. Following their return to the cytoplasm, GTP-hydrolysis, and RanGDP-release, they can import the next cargoes. Exportins (e.g., Exportin 1/Xpo1 a.k.a. CRM1) function similarly but recruit cargoes from the nucleus (along with RanGTP) and release them into the cytoplasm.

Each of such transport cycles transfers one Ran molecule from the nucleus to the cytoplasm. NTF2 (Nuclear transport factor 2) returns

[1]Department of Cellular Logistics, Max Planck Institute for Multidisciplinary Sciences, Göttingen, Germany. [2]Quantitative Biology, IRI Life Sciences, Humboldt-Universität zu Berlin, Berlin, Germany. [3]Department of Biological Optomechanics, Max Planck Institute for the Science of Light, Erlangen, Germany. ✉e-mail: goerlich@mpinat.mpg.de

RanGDP to the nucleus[13] and thus has to pass NPCs more frequently than any other NTR. A single NPC can accommodate up to 1000 facilitated events or a mass flow of 100 MDa per second[14], with transit times in the order of 10 ms for simple importin·cargo complexes[15,16] and ~25 ms for 60S pre-ribosomes[17,18]. NPCs have the capacity to transport many NTR·cargo complexes in parallel[19]. NPC passage per se is reversible and energy-independent. Active, directed transport requires additional energy input, e.g., by the aforementioned RanGTPase system.

FG nucleoporins (FG Nups; ~10 different ones in any given species) contain so-called FG repeat domains[20–22]. These domains comprise numerous FG (phenylalanine-glycine) motifs, are of low sequence complexity, and are intrinsically disordered[23,24]. FG domains are anchored to the inner scaffold of the NPCs[2,3] and bind NTRs during facilitated translocation[25,26]. Moreover, they can engage in multivalent, cohesive interactions via FG motifs and other hydrophobic groups to form a 'selective FG phase'[14,27–33].

State of matter similar to the FG phase was later described also as biomolecular condensates or liquid-liquid phase separation (LLPS)[34–41], e.g., in germline granules. Recently, it has become clear that multivalent interactions between interaction sites, called "stickers" drive phase separation and valence, strength, and patterning of stickers are key determinants of phase separation or properties of these condensates[39,42–44].

FG phases can easily be reconstituted and indeed behave like the permeability barrier of NPCs, i.e., they are a good solvent for NTRs and NTR·cargo complexes but exclude 'inert' macromolecules that lack FG interactions[28,29,32,45,46]. Transport through NPCs can thus be described as mobile species partitioning into this FG phase and exiting on the opposite side. Another perspective considers that multivalent FG contacts result in a 3D sieve whose mesh size determines the (passive) sieving properties of the phase. The binding of NTRs to FG motifs competes with FG-FG interactions and allows NTRs to 'melt' (along with their cargoes) through the meshes.

Traditionally, NPCs have been assumed to be sieve-like barriers with a 5 nm size limit for passive passage[11]. This corresponds to the diameter of GFP. Later, however, it became evident that there is actually a continuum between passive and facilitated NPC-passage[45,47]. Indeed, transport rates are not only determined by the size but also by surface properties of the mobile species, with negative charges or lysines impeding NPC passage, while "FG-philic" residues (like R, H, C, or hydrophobic ones) speed up. Based on these principles, GFP has been engineered to pass NPCs either very slowly (super-inert GFP variants) or very rapidly (GFP[NTR] variants), with the extreme versions (of identical size) differing more than 10,000-fold in their NPC passage rates[45].

High transport selectivity of an FG phase requires the local FG repeat concentration to exceed a threshold of ~200 mg/ml[28]. Such high local concentration can only partially be imposed by anchoring the ~15 MDa of FG domain mass[48] to the NPC scaffold. The other key contribution is localized phase separation, in particular, of Nup98 FG domains[32].

FG domains from Nup98 or its homologs[22,32,49] (e.g., Nup116, Nup100, and Nup145 in yeast) are the most cohesive[32,50] and crucial ones for maintaining the permeability barrier of NPCs[31,47,51,52]. They are typically dominated by GLFG motifs and comprise ~500–700 residues. Of all FG domains, they feature the highest number (~50) and density of FG or FG-like motifs (one per ~12 residues). They are extremely depleted of charged residues, experience water as a poor solvent, and readily phase-separate from low µM or even sub-µM concentrations to form condensed FG phases with 200–300 mg/ml or 100–300 mM FG repeat units[32]. Such self-assembled Nup98 FG phases recapitulate the transport selectivity of NPCs particularly well[32,45,46]. They exclude inert (i.e., "FG-phobic") macromolecules (such as the 25 kDa-sized mCherry) to partition coefficients below 0.05 while allowing rapid entry of NTRs

and NTR·cargo complexes. NTF2, for example, reaches a partition coefficient of ~2000. RanGTPase-controlled cargo import and export scenarios can also be recapitulated faithfully with a single Nup98 FG phase[46]. Therefore, Nup98 FG phases capture the essential features and represent simple experimental models of NPC-typical transport selectivity.

We now systematically tested the relevance of conserved Nup98 sequence features and found charge depletion and balanced hydrophobicity to be key parameters for the phase system. Introducing even a moderate negative charge into the repeats abolished phase separation. Surprisingly, however, there is no fundamental requirement for FG motifs. The GLFG→GLFS mutation, for example, preserved the assembly of a selective phase that excluded mCherry and still allowed a very efficient influx of all NTR·cargo species tested. Even a YG phase (with phenylalanines changed to tyrosines) displayed remarkable transport selectivity, e.g., for the engineered 3B7C GFP[NTR]-variant. YG motifs appear, however, rather incompatible with NTR-binding, suggesting co-evolutionary adaptations between wild-type FG domains and NTRs. We now matched this preference pattern of native NTRs with an advanced tetraGFP[NTR] variant that is particularly well adapted to GLFG motifs. Shrinking inter-FG spacers favoured phase separation, increased the intra-phase protein density, caused a stricter exclusion of FG-phobic species, and impeded the entry of NTRs carrying cargo. Extending spacers had the opposite effects, supporting the model of a sieve-like barrier whose mesh size is ruled by the distance of connection points. NTRs bind FSFG motifs well; however, mutating GLFG in the Nup98 FG domain to more hydrophobic FSFG motifs resulted in a hypercohesive phase repelling not only inert species but also precluding any NTR entry. Extending the inter-FSFG spacers relaxed this hypercohesive phenotype. This is consistent with a permeability barrier maintained by cohesive repeat contacts and with NTRs traversing the barrier by resolving such cohesive interactions locally.

## Results

### Experimental approach

Nup98 is central to the NPC permeability barrier. Its FG domain is highly cohesive and comes with several remarkable evolutionarily conserved features, such as depletion of charges, a dominance of GLFG motifs, and a rather high FG density with about eight motifs per 100 residues (Supplementary Table 1). Conservation implies evolutionary pressure and functional relevance, and so we wondered how changes in Nup98-typical features would impact the phase behaviour.

In order to characterize FG domain variants, we used codon-optimized expression vectors to produce them in *E. coli*, performed purifications under denaturing conditions, and prepared 1 mM FG domain stocks in 4 M guanidinium hydrochloride (to keep them initially non-interacting). Phase separation was then initiated by a 100-fold dilution in a guanidinium-free buffer[32,46]. This setup was used to determine saturation concentrations ($C_{sat}$) by measuring the fractions that remained soluble after pelleting the condensed FG phases. In separate assays, we used confocal laser scanning microscopy (CLSM) to determine how various fluorescent probes partition into the FG phases.

### A simplified FG domain as a starting point

Our starting point was the FG domain from *Tetrahymena thermophila* MacNup98A (Fig. 1a) because it is already well characterized and avoids complications such as an elevated amyloid propensity or a need for O-GlcNAc modifications[50]. For conceptual clarity, we used a derivative that had been simplified[46], namely by (1) converting all FG and FG-like motifs to GLFG motifs and (2) shifting the GLFG motifs to equidistant positions in the sequence. This simplified sequence contains 52 GLFG motifs in total (each separated by a spacer of eight

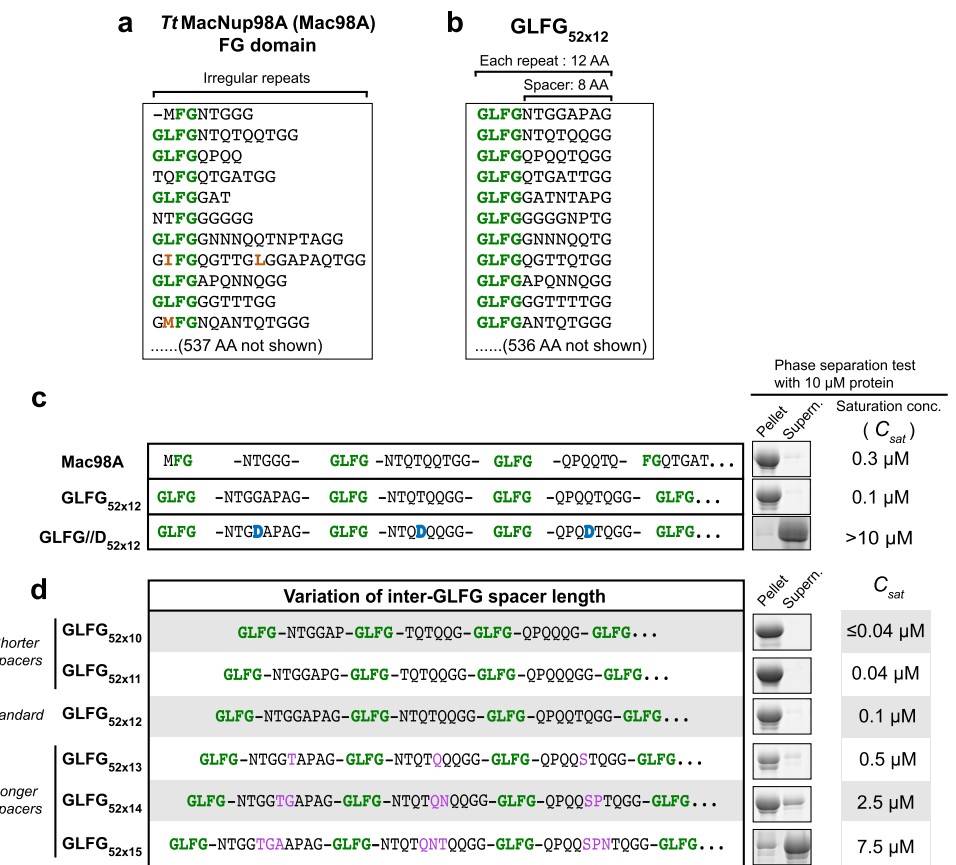

**Fig. 1 | Inter-GLFG spacer length determines the phase separation propensity.**
**a**, **b** Sequences (partial) of the wild-type *Tetrahymena thermophila* MacNup98A ("Mac98A") FG domain (**a**) and a sequence-regularized variant, GLFG$_{52\times12}$ (**b**). The latter served as a template for variant design in this study. GLFG$_{52\times12}$ is composed of 52 repeat units, each containing one GLFG motif (green) and one eight-amino-acid spacer (length of each repeat unit = 12 AA). For space economy, only the N-terminal -130 residues of each are shown (see Supplementary Note 1 for complete sequences). **c** Based on GLFG$_{52\times12}$, a variant with one residue in each inter-GLFG spacer replaced by Asp was constructed (GLFG//D$_{52\times12}$). The sequence is listed with that of Mac98A and GLFG$_{52\times12}$. For each, the N-terminal sequence up to the fourth FG motif is shown. For GLFG$_{52\times12}$ and GLFG//D$_{52\times12}$, the C-terminal sequences follow the same design strategy as the N-terminal sequences shown (see Supplementary Note 1 for complete sequences). Each FG domain or variant was expressed, purified, and dissolved at a concentration of 1 mM in 4 M guanidinium hydrochloride. As a test of phase separation property, this stock of each was diluted 100-fold quickly with assay buffer (50 mM Tris/HCl pH 7.5, 150 mM NaCl, 5 mM DTT) to allow phase separation. The dilution was centrifuged. SDS samples of the obtained pellets (FG phase), if there were, and supernatants (soluble content) were loaded for SDS-PAGE at an equal ratio, followed by Coomassie blue-staining for quantification. Saturation concentration ($C_{sat}$) of each was taken as the concentration of the supernatant. For GLFG//D$_{52\times12}$, no phase separation was detected at the assay protein concentration (=10 μM), and the exact $C_{sat}$ was not determined. **d** Based on GLFG$_{52\times12}$, variants with the same number (=52) of GLFG motifs but varying inter-GLFG spacer lengths were constructed. N-terminal sequences up to the fourth FG motif are shown (see Supplementary Note 1 for complete sequences). For each variant, a 1 mM stock in 4 M guanidinium hydrochloride was prepared, and phase separation was analysed as described above. **c**, **d** Each of the assays was performed twice on independent samples with similar results, and representative images are shown.

variable amino acids); it has a repeat length of 12 residues, containing one GLFG motif per 12 residues, and is referred to as "GLFG$_{52\times12}$" (Fig. 1b). We tried to preserve the original overall hydrophobicity and amino acid composition and kept the intervening Gle2-binding sequence (GLEBS) unchanged (see Supplementary Note 1 for complete sequences of the FG domains/variants). The simplification had only marginal effects on the phase separation properties and the transport selectivity of the resulting FG phase (Figs. 1–3; and ref. [46]), suggesting that GLFG$_{52\times12}$ captures the barrier properties of the original domain indeed very well.

### A negative net charge antagonizes FG phase separation

We first tested whether the observed uncharged nature of Nup98 FG repeats is relevant and observed that introducing one negative charge (as an aspartate) per repeat completely abolished phase separation (see GLFG//D$_{52\times12}$ variant in Fig. 1c). This can be rationalized by electrostatic repulsion and thus an increase in water solubility of the spacers and also the entire domain[53,54]. It is consistent with the extreme selection against Asp and Glu in native Nup98 FG domains and other

cohesive FG domains[55,56], and against a negative net charge in FG domains in general.

### Shorter inter-FG spacers enhance cohesive interactions

In the next step, we varied the length of the inter GLFG-spacers while keeping their amino acid composition and the number of GLFG motifs constant (Fig. 1d). For shorter-spacer variants (GLFG$_{52\times10}$ and GLFG$_{52\times11}$), residues were deleted such that the remaining spacers matched the original composition as closely as possible (Supplementary Table 2). Likewise, amino acid insertions to extend the spacers (GLFG$_{52\times13}$, GLFG$_{52\times14}$, and GLFG$_{52\times15}$) were chosen according to the same constraint.

These spacer length variations had a striking effect on phase separation, with shortening by one residue (per spacer) decreasing $C_{sat}$ by a factor of ~3–5 and thus favouring phase separation. $C_{sat}$ of GLFG$_{52\times10}$ (with six-amino-acid spacers between the GLFG motifs) was so low that it could hardly be measured by the pelleting assay. Conversely, extending the spacers increased the saturation concentration and thus disfavoured phase separation.

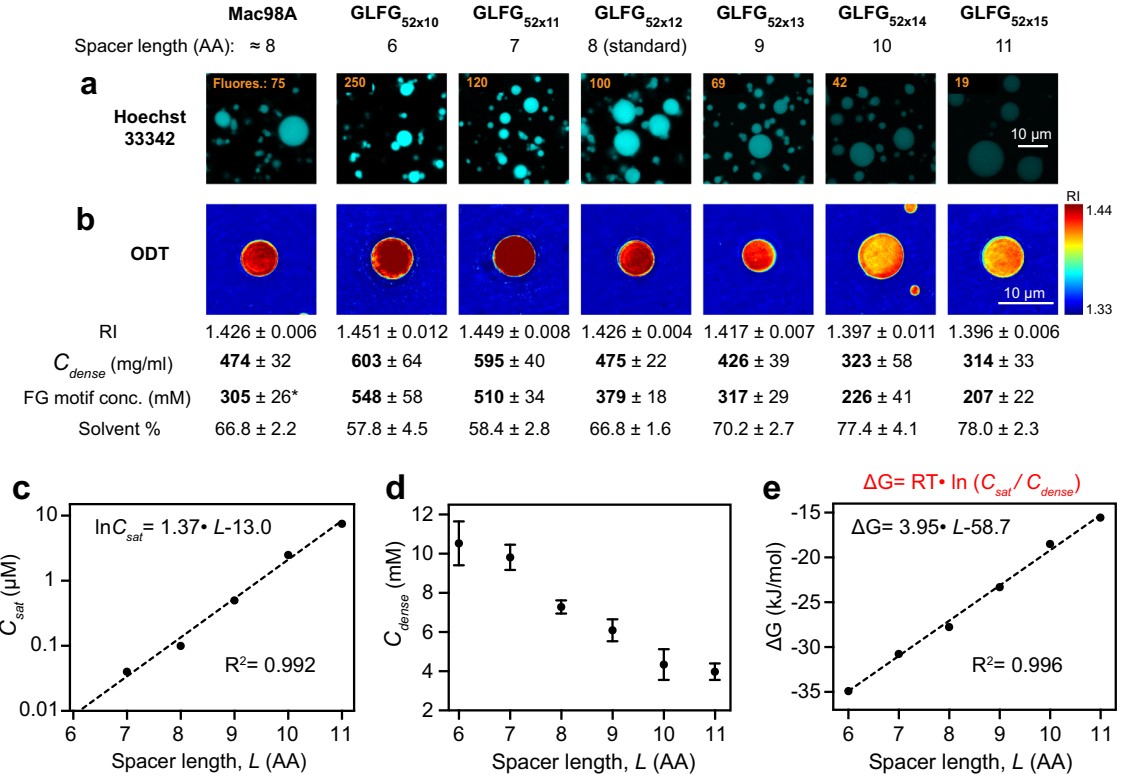

**Fig. 2 | Variation of inter-GLFG spacer length impacts intra-FG phase density.**
**a** Mac98A FG domain and indicated variants (the former has irregular spacer length with an average of about 8 AA), were dissolved at a concentration of 1 mM (except that GLFG$_{52×15}$ was dissolved at a concentration of 2 mM, because of its higher saturation concentration) in 4 M guanidinium hydrochloride, and phase separation was initiated by a rapid 50-fold dilution with assay buffer. Each was further diluted fourfold in 12 µM Hoechst 33342. Samples were analysed by confocal laser-scanning microscopy (CLSM). The numbers in orange indicate the fluorescence intensities of the Hoechst dye inside the FG phases. The fluorescence intensities were relative to that of GLFG$_{52×12}$ (arbitrarily set to 100). The above assay was performed twice on independent samples with similar results, and representative images are shown. **b** Phase separation of Mac98A FG domain and its variants was initiated similarly as in **a** and the samples were analysed by optical diffraction tomography (ODT) independently (in the absence of Hoechst). Panels show maps of refractive index (RI). For each, ten independent FG particles were analysed. Representative images and mean values (±S.D. between the ten FG particles) of RI, mass density ($C_{dense}$), FG motif concentration and solvent % are shown. *Note: Mac98A FG domain has <52 FG motifs but also contains FG-like motifs/ hydrophobic residues in spacers (Supplementary Table 2), which are not counted here. **c** $C_{sat}$ shown in Fig. 1d is plotted against the spacer length (L) of variants. Mean values of $C_{sat}$ (n = 2) are plotted and fitted to a simple exponential function (dashed line) with the R-squared value indicated. **d** $C_{dense}$ obtained from ODT is plotted against L. For each, ten independent FG particles were analysed, and data are presented as mean ± S.D. (in molar concentration). **e** Gibbs free energy for phase separation (ΔG) of each variant was calculated by the equation in red (Eq.1 in the main text; with T = 298 K) and is plotted against L. The data are presented as mean values. Note: $C_{sat}$ of GLFG$_{52×10}$ was extrapolated from the equation derived in **c** (=0.008 µM), and the corresponding ΔG was computed accordingly.

Above their saturation concentrations, all FG domains of the spacer length series assembled into µm-sized, near-spherical FG phases ("FG particles") that stained bright with Hoechst 33342 (Fig. 2a). This dye is environmentally sensitive and shows a greatly enhanced fluorescence upon shielding from water. It is best known as a DNA-specific probe[57,58] but also stains FG phases in a way that depends on the presence of cohesive interactions[46]. Indeed, the Hoechst signals in Fig. 2a correlate clearly with the cohesiveness of the FG domains: the variant with the shortest spacers and the lowest $C_{sat}$ (GLFG$_{52×10}$) stained 2.5 times brighter than the standard GLFG$_{52×12}$ phase and ~13 times brighter than the least cohesive one (GLFG$_{52×15}$). Thus, the local FG motif concentration in the phase and/or occupancy of cohesive interactions appear to increase with shorter spacers.

To determine the protein content of the phases ($C_{dense}$), we applied optical diffraction tomography (ODT), which exploits the refractive index (RI) as a robust concentration measure (Fig. 2b, d; refs. [59–61]). This way, we obtained a $C_{dense}$ of ~470 mg/ml for the original Mac98A and the standard GLFG$_{52×12}$ variant. This number is higher than previous estimates based on signal-matching between Alexa488-labelled FG domains and a dilute phase marker[32]; but some fluorescence quenching within such a dense phase might explain this difference. The protein concentration of 470 mg/ml (380 mM FG motifs in

the GLFG$_{52×12}$ phase) leaves a solvent content of ~67% (assuming a partial specific volume of 0.7 ml per gram of FG domain). Shrinking the spacers between the GLFG motifs from 8 to 6 residues (in GLFG$_{52×10}$) increased $C_{dense}$ considerably to ~600 mg/ml (550 mM FG motifs), leaving a solvent content of 58%. Conversely, extending the spacers to 11 residues decreased the FG domain concentration to ~310 mg/ml (210 mM FG motifs), leaving 78% solvent.

There was a remarkable log-linear relationship between $C_{sat}$ (Fig. 2c) or the $C_{sat}$:$C_{dense}$ ratio and the spacer length. On the other hand, the $C_{sat}$:$C_{dense}$ ratio is also linearly related to the free energy change of phase separation[62]:

$$\triangle G = RT \bullet \ln\left(\frac{C_{sat}}{C_{dense}}\right) \qquad (1)$$

where R is the gas constant (8.31 J/mol K) and T the absolute temperature in Kelvin. ΔG thus depends inversely on the spacer length (Fig. 2e). Perhaps, shorter spacers entropically favour hydrophobic interactions of the LF clusters[63,64] and/or enhance the local cooperativity of such cohesive contacts[53,54]. The effect can also be interpreted as short spacers increasing the overall hydrophobicity of the FG domain and thus decreasing its water solubility.

## Shorter spacers make the FG phase a stricter barrier

Nup98 FG phases provide transport selectivity because they can exclude non-interacting ("FG-phobic") macromolecules, thereby impeding their passage and thus controlling fluxes through NPCs. One mechanistic interpretation is that the phase-entry of mobile species requires a local disruption of cohesive interactions, which imposes a thermodynamic penalty. By providing a spectrum of cohesion strengths, the spacer length series now allowed to explicitly test this concept.

For a first test (Fig. 3), we used four monomeric fluorescent proteins (of ~30 kDa) as probes, whose surfaces ranged from very inert/FG-phobic to weakly FG-philic (mCherry, EGFP, efGFP_8Q, efGFP_8R; ref.[45]). The GLFG$_{52\times10}$ phase with the shortest spacers excluded all four proteins very well (with partition coefficients of ≤0.05–0.07). The spacer extension increased their partitioning within the GLFG$_{52\times15}$ phase, namely to 0.12 (for mCherry) or even to 1.5 (for efGFP_8R). Thus, less cohesion appears to lower the energetic barrier for phase entry.

We next asked whether the partitioning of NTRs is also sensitive to spacer length and observed that NTF2 (coupled with Alexa488 for tracking) partitioned in all FG phases equally well with very high partition coefficients between 2700 and 3300. This contrasts the behaviours of the just mentioned fluorescent proteins but can be explained by NTF2 engaging in more FG contacts when immersed in a phase with a higher FG density. We would assume that the entire NTF2 surface is available for such interactions and provides a sufficient number of binding sites to respond to the higher FG density.

Indeed, we observed a rather different trend when NTF2 was complexed with its cognate cargo, RanDP (in this case, Ran was labelled with Atto488). The NTF2·Ran complex partitioned well (with coefficients > 200) in the GLFG$_{52\times12}$ phase and any phase with longer spacers; however, it arrested at the surface of the GLFG$_{52\times10}$ phase and essentially failed to enter. This is the expected phenotype when the FG-philic NTF2 immerses into the phase while Ran still faces the bulk solvent.

Next, we tested three additional NTR·cargo complexes, namely (1) transportin·M9-EGFP (with M9 being the nuclear import signal of hnRNP A1[65]), (2) importin β·IBB-EGFP (with IBB being the Importin Beta-Binding domain from importin α—a very strong import signal[66]), and (3) the exportin Xpo1 bound to RanGTP and NES-EGFP (with the NES being a PKI Nuclear Export Signal[67]). All three complexes were arrested at the surface of the short-spacer GLFG$_{52\times10}$ phase but efficiently entered the long-spacer phase variants.

The inefficient entry of these species into the short-spacer FG phases could be explained by their larger sizes (53–170 kDa) as compared to NTF2 alone (30 kDa). An alternative explanation is that entry is impeded by larger FG-phobic surface regions of Ran and/or EGFP. Indeed, the partitioning of the importin β·IBB-fusion into the GLFG$_{52\times10}$ or GLFG$_{52\times11}$ phases increased 30 or 1000-fold, respectively, when EGFP was exchanged for sffrGFP7$^{NTR}$, a GFP derivative engineered for an FG-philic surface[45]. Likewise, the tetrameric (110 kDa) GFP$^{NTR}$_3B7C, which was also engineered for FG-philicity and rapid NPC passage[45], partitioned very well in all FG phases. Therefore, we assume that shorter spacers not only increase cohesion but, in consequence, also cause a stricter exclusion of macromolecules with inert surfaces. In line with this, such a strict selectivity can also be relaxed by just lowering the salt concentration in the assay buffer, which weakens hydrophobic inter-repeat contacts (Supplementary Fig. 1; refs. [33,62]). Increasing the salt concentration had the opposite effect, in particular for the long-spacer FG phases. These data show that selectivity, in general, depends on the strength of cohesive interactions in the FG phase.

## Phase separation tolerates FG motif mutations if overall hydrophobicity is preserved

We next analysed a series of GLFG motif mutations, in the context of the standard-spacer GLFG$_{52\times12}$ variant. In line with previous observations[27,31,33,55,68], a reduction in the motif hydrophobicity essentially abolished phase assembly (Fig. 4a). This applied to the GLFG→GAFG mutation (Leu replaced by Ala), the similar GLFG→GAYG mutation, as well as to the GLFG→GLLG mutation (Phe changed to a less hydrophobic Leu). Phase separation of the latter was, however, restored when a spacer residue was mutated to a more hydrophobic leucine (GLLG//L$_{52\times12}$). Note that the double mutation is roughly neutral with respect to the count of hydrophobic (aromatic+ aliphatic) carbons per repeat unit.

Likewise, shifting the leucine from the GLFG motif to the spacer without altering the amino acid composition (GXFG//L$_{52\times12}$) preserved phase separation. In addition, we tested several additional motifs (GFLG, GLYG, GIFG, GLFA, SLFG, and GLFS) with similar hydrophobicity as the GLFG motif and observed phase separation as well. The protein concentration in the resulting dense phases was also remarkably similar (Fig. 4b, c). Thus, phase separation tolerates deviations from the GLFG motif sequence as long as the overall hydrophobicity is not reduced. This is consistent with the cohesive behaviours of FG domains that lack GLFG motifs but are sufficiently hydrophobic (e.g., Nup42, the N-terminal Nsp1, or the Nup62/54/58 FG domains; refs. [50,55,56,68,69]).

Changing GLFG to GLYG motifs lowered $C_{sat}$ below the detection limit of our assay. This is in line with the higher interaction strength of Tyr than Phe, as reported in other systems[70–73].

## FG-like phases can show high transport selectivity even without FG motifs or phenylalanines

We then tested all the above-mentioned FG phases (with GLFG, SLFG, GIFG, GXFG//L, or SLFG motifs) and FG-like phases (with GFLG, GLFA, GLFS, GLYG, or GLLG//L motifs) in permeation assays. All of them excluded mCherry (27 kDa) very well (with partition coefficients of <0.05 to 0.1) but allowed the accumulation of the tetrameric (110 kDa) GFP$^{NTR}$_3B7C variant to partition coefficients of 200 to 2900 (Figs. 5 and 6). This even applied to repeats where the FG motifs were replaced by YG motifs (GLYG$_{52\times12}$) or repeats lacking Phe and Tyr altogether (GLLG//L$_{52\times12}$). FG- and FG-like phases can thus show very high transport selectivity—independently of which hydrophobic motif confers the cohesive interaction. The engineered GFP$^{NTR}$_3B7C variant appears here to be a rather promiscuous client of these phases.

## Effects of FG motif mutations on phase entry of NTRs

We then tested how various NTRs and NTR·cargo complexes partition into our set of FG and FG-like phases. Mutating the GLFG to GIFG motifs (i.e., Leu replaced by Ile) had no noticeable effect on transport selectivity (Fig. 5). Given that leucine and isoleucine are very similar amino acids, this exchange appears indeed rather subtle. The change to SLFG motifs also had no detrimental effect on NTR entry but increased the partition coefficients for most NTRs and NTR·cargo complexes (up to four-fold). This is consistent with SLFG motifs being rather common, e.g., in the wild-type Nup98 FG domain from *C. elegans* (Supplementary Table 1), and that the slightly weaker cohesion of this motif (as measured by phase separation; Fig. 4) might promote NTR influx (see also below).

The GLFG→GLFS mutation marks a departure from FG motifs. Indeed, GLFS motifs are very rare in Nup98 FG domains (Supplementary Table 1). Nevertheless, it had a surprisingly small effect: just a ~threefold lower partition of NTF2 and ~twofold higher of importin β, transportin, and Xpo1 complexes.

Other mutations, however, were very detrimental to NTR interactions (Fig. 6): The GFLG mutation (with swapped Phe and Leu positions) reduced phase entry of importin β and the exportin Xpo1 drastically (20- to 200-fold). The GXFG//L mutation (Fig. 4) was created by swapping the leucine of the GLFG motif with a residue of the spacer (X = G, N, Q, S, T, A, P…). This reduced the influx of the importin β and transportin·cargo complexes nearly to the extent of phase-exclusion—

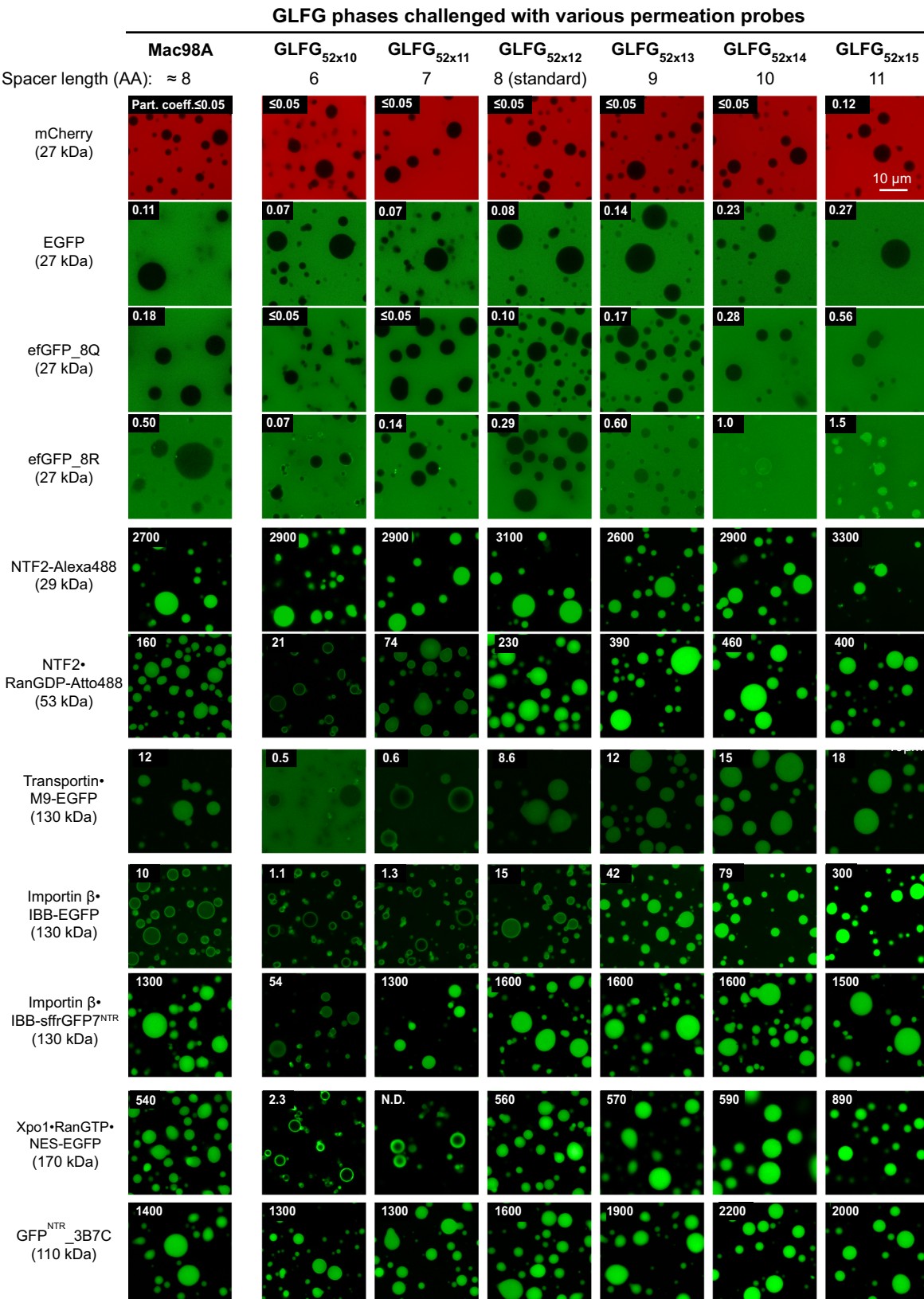

**Fig. 3 | Variation of inter-GLFG spacer length impacts the transport selectivity of barrier.** FG phases assembled from the Mac98A FG domain and indicated variants were prepared as described in Fig. 2. They were challenged with the indicated permeation probes (for probe descriptions, see main text). In each case, the calculated molecular weight of the complex is indicated. Scanning settings/ image brightness were adjusted individually to cover the large range of signals. The numbers in white refer to the partition coefficients of the fluorescent species into the FG phases (fluorescence ratios in the central regions of the particles to that in the surrounding buffer). N.D.: Not determined due to difficulties in defining the central region. Each of the above assays was performed twice on independent samples with similar results, and representative images/mean values are shown. The same applies to Figs. 5–10.

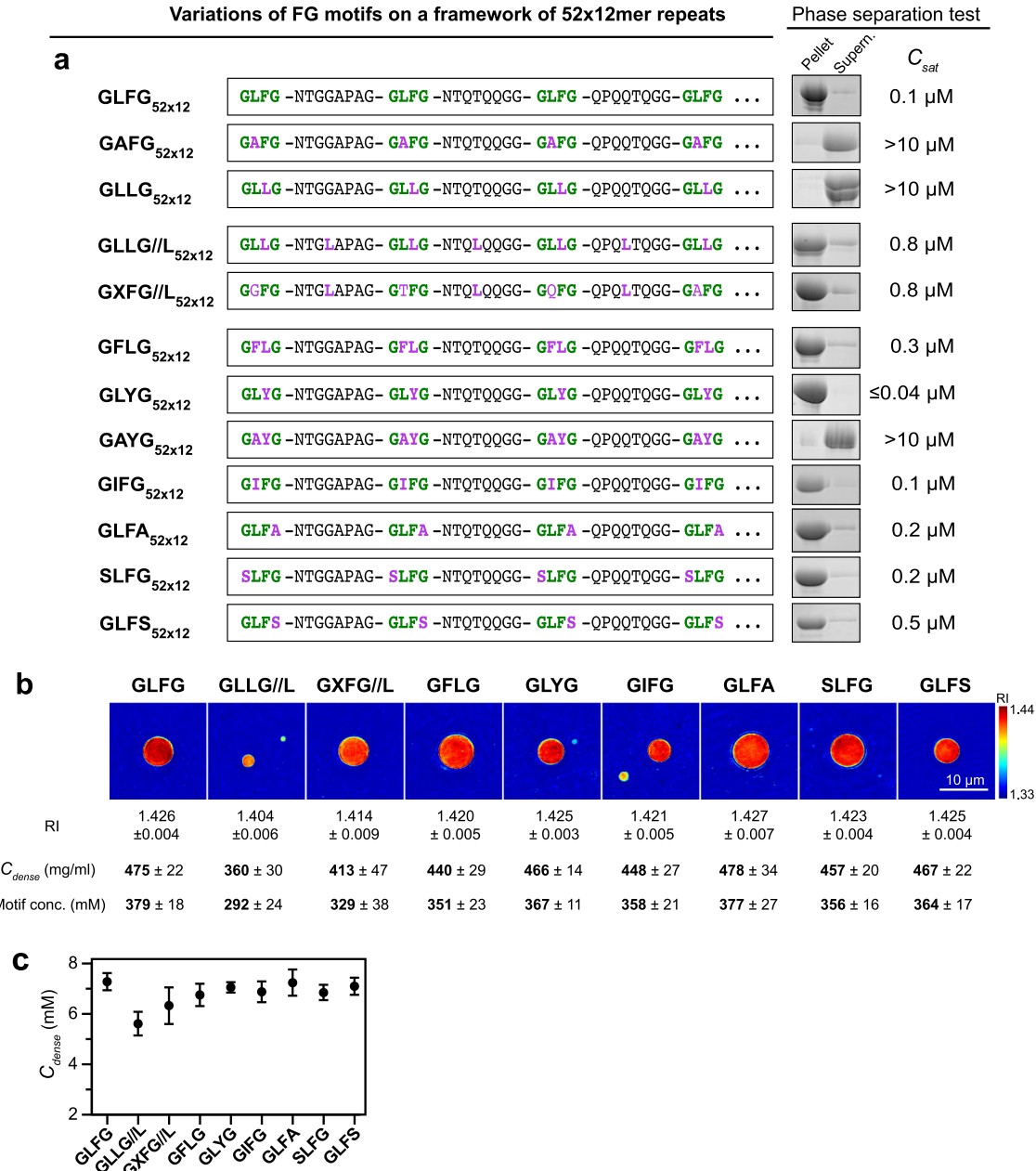

**Fig. 4 | Phase separation property is dependent on the overall hydrophobicity.**
**a** $GLFG_{52\times12}$ serves as a template for variant design in this experiment. Canonical FG motifs are coloured green and mutations are coloured purple. Note that the inter-GLFG spacers of $GLFG_{52\times12}$ do not contain Phe and Leu. All the variants contain 52 repeat units, and each unit is 12-amino-acid long. The N-terminal sequence of each variant up to the fourth FG motif is shown; the rest of the sequence (~620 residues/ 48 repeat units) follows the same design strategy as shown (see Supplementary Note 1 for complete sequences). Phase separation of the variants was analysed as in Fig. 1c at [Variant] = 10 µM. No phase separation was observed for $GAFG_{52\times12}$, $GLLG_{52\times12}$, and $GAYG_{52\times12}$ under these conditions, and the exact saturation concentrations were not determined. Each of the assays was performed twice on independent samples with similar results, and representative images are shown. **b** Phase separation of $GLFG_{52\times12}$ and its variants was initiated and the resulting dense phases were analysed by optical diffraction tomography (ODT). Panels show maps of refractive index (RI). For each, ten independent FG particles were analysed. Representative images and mean values (±S.D. between the ten FG particles) of RI, mass density ($C_{dense}$) and motif concentration are shown. **c** $C_{dense}$ for each FG phase variant determined by ODT. For each, ten independent particles were analysed, and the data are presented as mean ± S.D. between the ten FG particles (in molar concentration).

though it still allowed a quite efficient accumulation of NTF2 and the Xpo1 complex. The tyrosine GLYG mutant, as well as the also phenylalanine-free GLLG//L mutant, essentially excluded importin β, transportin, and Xpo1 complexes. Likewise, phase entry of NTF2 and the NTF2·RanGDP complex was 50-fold reduced. These observations suggest co-evolutionary adaptations between FG domains and FG-binding sites in native NTRs, with FG-binding pockets[74–77] being probably more constrained than surface sites.

## Arginines promote FG phase entry not just through cation·π interactions

Lysine and arginine are very similar amino acids with positively charged sidechains. On the surface of a mobile species, however, the two have profoundly different effects. Exposed lysines render a mobile species FG-phobic and slow in NPC passage. By contrast, arginines favour partitioning in the FG phase and can speed up NPC passage by a large factor[45]. The $GLFG_{52\times12}$ phase recapitulated this behaviour

## FG phase variants challenged with various permeation probes

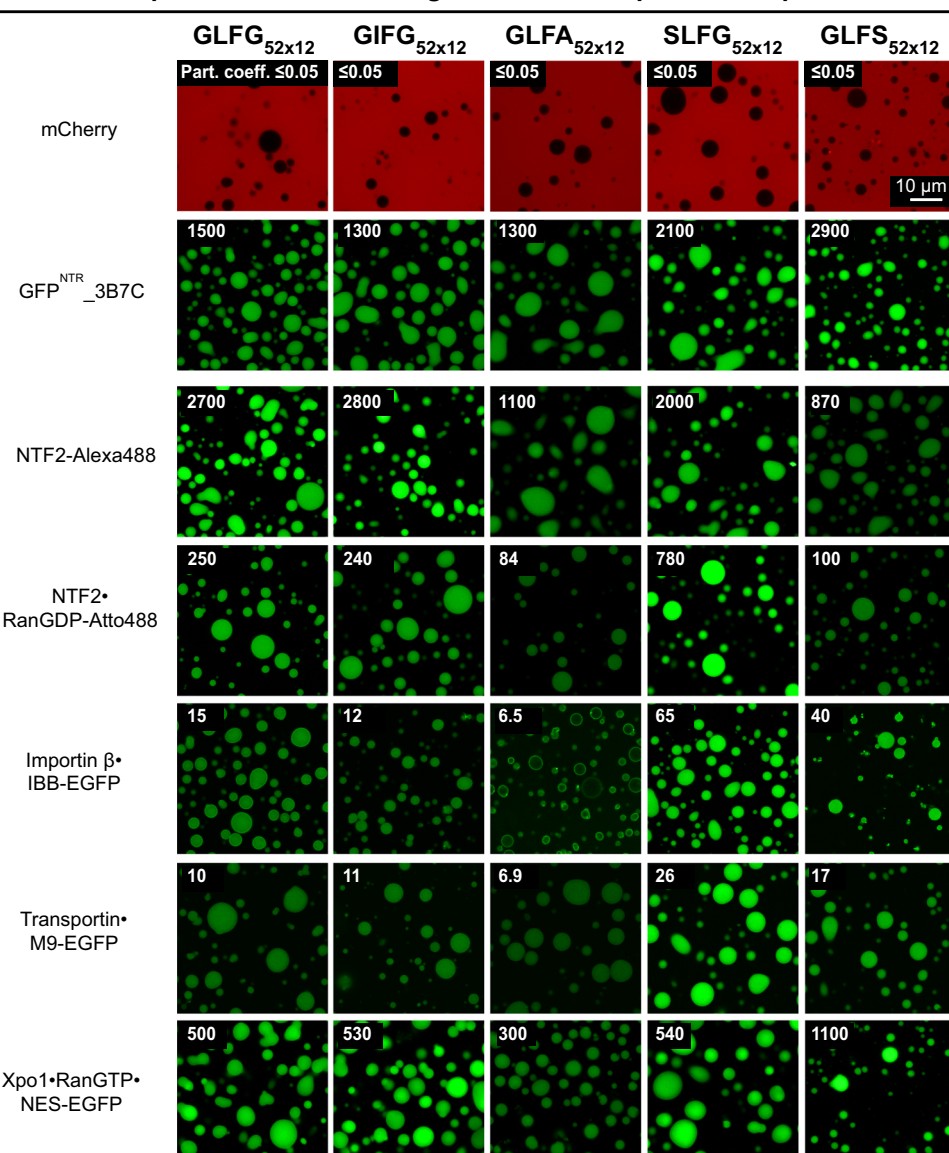

**Fig. 5 | FG domain variants with non-canonical Fx motifs phase-separate into barriers that allow entry of NTR-cargo complexes.** FG or FG-like phases assembled from the indicated variants were challenged with the indicated probes. Scanning settings/ image brightness were adjusted individually due to the large range of signals. The numbers in white refer to the partition coefficients of the fluorescent species into the phases.

(Fig. 7). It accumulated the sffrGFP4$^{NTR}$ variant (with 25 solvent-accessible arginines and no lysines) to a partition coefficient of ~20, but it excluded the sffrGFP4 25 R→K variant (with 25 solvent-exposed lysines instead) down to a partition coefficient of 0.1. The enhancing effect of arginines can be explained by the positively charged guanidinium group engaging in cation-π interactions with the π electron cloud of phenylalanines[78]. The multiplicity of arginines amplifies the effect to a ~200-fold difference in partition coefficient between the two GFP variants.

The GLYG phase also accumulated sffrGFP4 while excluding the 25 R→K variant. This was expected—given that tyrosine is also known for forming similar cation-π interactions with arginine. The GLLG//L phase, however, contains neither phenylalanines nor tyrosines, and yet it accumulated sffrGFP4 80-fold higher (to a partition coefficient of 8) than the 25 R→K variant (of 0.1). This unexpected observation strongly argues for functionally relevant arginine interactions that are distinct from cation-π (see below).

### TetraGFP$^{NTR}$: an engineered FG (GLFG) phase marker

We previously described the GFP$^{NTR}$_3B7C variant, which was engineered to partition particularly well into FG phases, to cross NPCs rapidly, and to provide a specific NPC stain when incubated with digitonin-permeabilised HeLa cells[45]. We used the X-ray structure of this homotetramer for Rosetta optimization and introduced a (buried) M225F mutation to stabilize the interface between the subunits. We refer to the new variant as TetraGFP$^{NTR}$. The mutation also made the probe more similar to native NTRs, namely by selectively suppressing the partitioning of TetraGFP$^{NTR}$ into the GFLG, GXFG, and GLYG phases while maintaining the very high accumulation in the GLFG phase (Fig. 8a). This enhanced specificity is also evident from NPCs-stains in digitonin-permeabilised cells, where TetraGFP$^{NTR}$ gave a crisp signal with even less background than its progenitor 3B7C (Fig. 8b, c). Moreover, TetraGFP$^{NTR}$ showed a preference for GLFG over the FSFG phase (see the section below). How the mutation introduced at the tetramer interface translates to enhanced GLFG selectivity is still

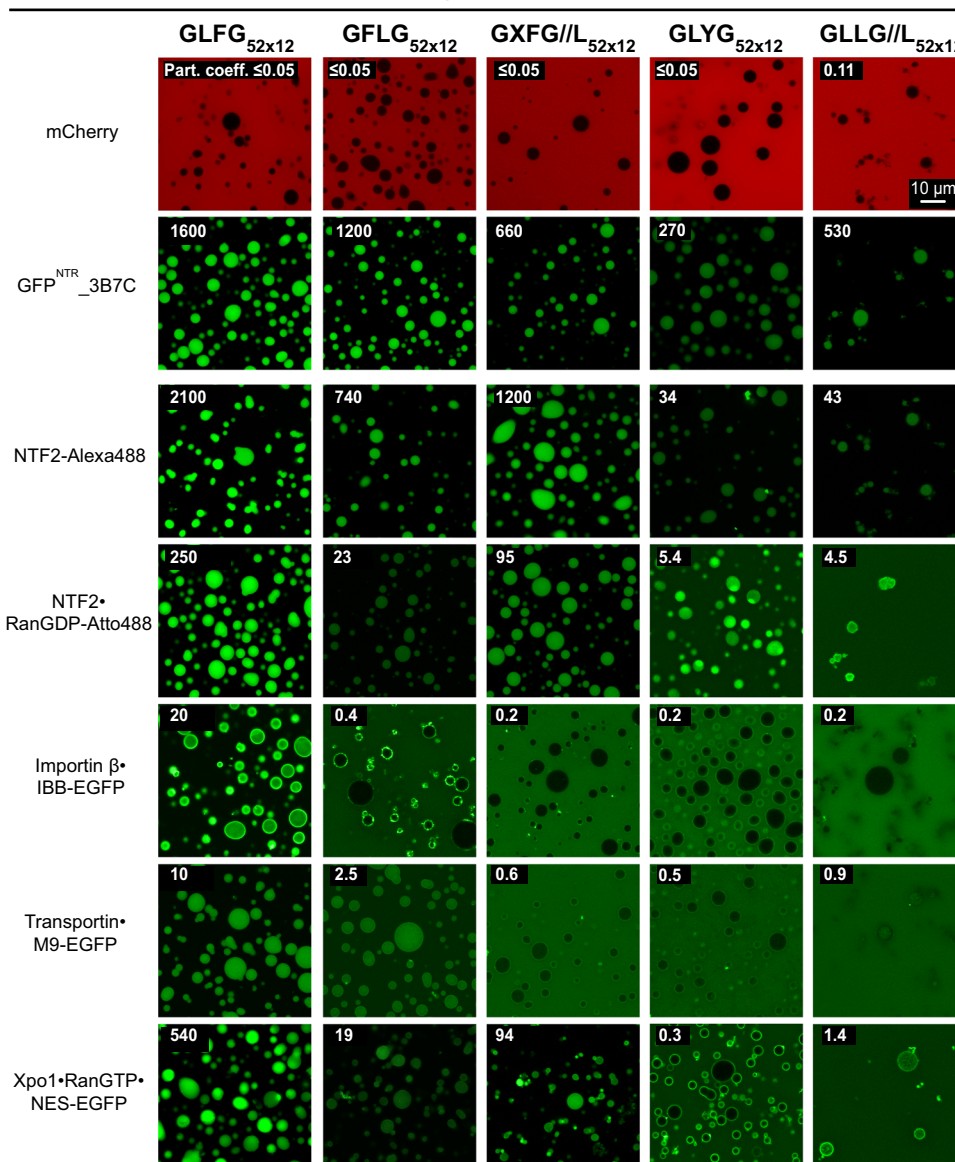

**Fig. 6 | Spatial arrangement of Phe and Leu in FG domain is crucial for NTR-binding but not for engineered GFP^NTR-variants.** FG or FG-like phases assembled from the indicated variants were challenged with the indicated probes. Scanning settings/ image brightness were adjusted individually due to the large range of signals. The numbers in white refer to the partition coefficients of the fluorescent species into the phases.

unclear. However, TetraGFP^NTR can serve as a very useful phase marker and translocation probe.

### Changing GLFG to FSFG motifs results in hyper-cohesiveness

GLFG motifs are dominant in Nup98 domains and their homologs, like the yeast Nup116 FG domain. This is, however, different in other Nup FG domains, an example being yeast Nsp1[20]. Its regular sub-domain (residues 274–601) contains exclusively (more hydrophobic) FSFG motifs separated by rather long, highly charged, and thus anti-cohesive spacers[69].

We aimed to compare FSFG and GLFG motifs directly and mutated all GLFG motifs in the GLFG$_{52\times12}$ domain to FSFG motifs (Fig. 9a). The FSFG$_{52\times12}$ domain phase-separated very well and perfectly excluded mCherry, indicating that it forms an effective barrier. NTRs bound to the surface but failed to reach the interior of the phase. This applied to all NTR species and, strikingly, even to cargo-free NTF2 and to GFP^NTR_3B7C, which had entered the above-described, very strict GLFG$_{52\times10}$ phase very well.

Two possible explanations for this phase behaviour came to our mind. First, the cohesion between FSFG motifs might be so strong that NTRs cannot pass these contacts. Second, the FSFG motifs are poor NTR binders when brought into the context of the uncharged Nup98 inter-FG spacers. To distinguish between these scenarios, we mutated (in two variants) one-third of the FSFG motifs to SSSG motifs (Fig. 9b, c). This lowered the FG density and reduced the overall hydrophobicity approximately to the level of the original GLFG domain. These partial F→S mutants indeed allowed a very efficient accumulation of all tested NTR species, with the importin β·IBB-EGFP complex reaching a partition coefficient of 200–300, NTF2 of ~2000, and 3B7C of ~1000. As the remaining FSFG motifs remained in their initial context, we can conclude that they are fully proficient in NTR-binding, also in the context of the non-charged Nup98 spacers. In turn, this suggests that the cohesion between FG motifs of the FSFG$_{52\times12}$ domain is so strong that even the otherwise productive FG·NTR interactions cannot disengage them.

## Antagonism between FG cohesion and NTR·FG interactions

The very strong cohesion of the FSFG$_{52×12}$ domain is also evident from its extreme phase separation propensity. $C_{sat}$ here is so low that the soluble fraction remained undetectable in our phase separation experiments. $C_{sat}$ decreases exponentially with the number of repeats[62], and even when reducing the number of FSFG$_{x12}$ repeats from 52 to 29 (FSFG$_{29×12}$, Fig. 9d), $C_{sat}$ (0.2 μM) was just above the detection limit. The very high protein concentration (650 mg/ml) in the FSFG$_{52×12}$ phase and the very strong Hoechst signal (Fig. 10b) are also consistent with extreme cohesion and high occupancy of inter-repeat contacts.

### FG phase variants challenged with GFP variants

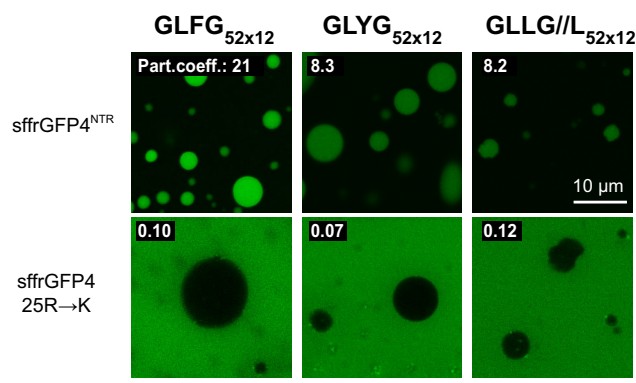

**Fig. 7 | Arginines promote FG phase entry not just through cation-π interactions.** Indicated condensed phases were challenged with sffrGFP4, which is an engineered FG-philic GFP variant, and "sffrGFP4 25 R→K", in which all the surface FG-philic arginine residues were replaced by FG-phobic lysines. Note that the GLLG//L phase, which lacks aromatic residues, still allows the partition of the Arg-rich sffrGFP4, indicating interactions other than cation-π promoted the entry.

In the next step, we asked if an NTR-permissive FSFG phase can also be generated with regular repeats. For this, we extended the inter-FSFG spacer length of the FSFG$_{52×12}$ domain from initially 8 to 11 residues (FSFG$_{52×15}$) or to 14 residues (FSFG$_{52×18}$; Fig. 10a). Indeed, the resulting dense phases dropped in protein concentration (to 530 and 410 mg/ml, respectively) and in Hoechst signal (7 and 60-fold, respectively). This came with relaxed barrier properties: The FSFG$_{52×15}$ phase accumulated NTF2 and 3B7C to partition coefficients of 2600 and 1300, respectively. Extending the spacer length to 14 residues (FSFG$_{52×18}$) then allowed also the importin β·IBB-EGFP complex to reach an intra-phase partition coefficient of 90. The hypercohesive nature of the FSFG motifs (Fig. 10b and Supplementary Fig. 2) can thus be balanced by longer inter-FG spacers that lower the cohesion propensity, probably through entropic effects. In any case, this set of experiments illustrates nicely the antagonism between inter FG-cohesion and FG·NTR interactions and suggests that selective NPC passage indeed happens by NTRs transiently disengaging cohesive contacts within the FG phase.

## Discussion

### Conserved features of Nup98 FG domains

Nup98 FG repeat domains readily phase-separate to form an FG phase with NPC-typical transport selectivity. They are intrinsically disordered, but in contrast to typical intrinsically disordered domains, they show a high degree of sequence conservation[32]. In fact, the Nup98 FG domain of vertebrates is as conserved as its globular anchor domain. This points to considerable evolutionary pressure to maintain its interactions. Features of Nup98 FG domains, which are conserved across all eukaryotic clades, include (1) an overall length of about 500–700 residues, (2) a low complexity sequence with an extreme selection against charged residues, against negatively charged ones in particular, but also against tyrosines, histidines, cysteines, tryptophans, valines or isoleucines, (3) a rather constant hydrophobicity with a density of one FG motif per 12 residues, and (4) a prevalence of GLFG motifs.

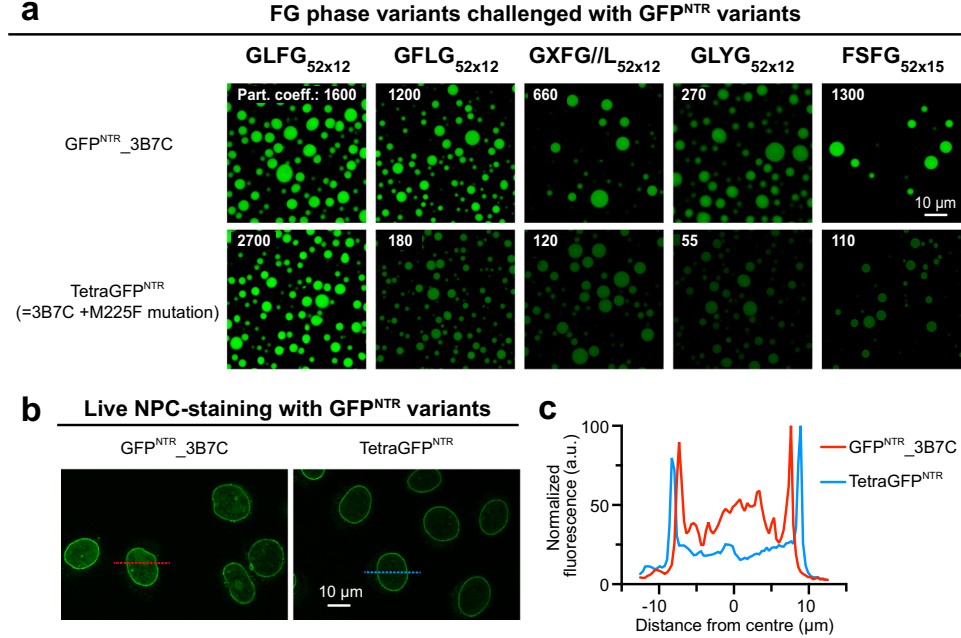

**Fig. 8 | An engineered GFP variant that binds specifically to GLFG motifs and NPCs. a** A rational mutation (M225F) was introduced to GFP$^{NTR}$_3B7C, and the resulting variant was named TetraGFP$^{NTR}$. Indicated FG or FG-like phases were challenged with GFP$^{NTR}$_3B7C and TetraGFP$^{NTR}$. Note that except for the canonical GLFG phase, the mutation reduces the partition coefficients into all other phases. **b** Panels show confocal scans of digitonin-permeabilized HeLa cells that had been incubated for 5 min with 250 nM of either GFP$^{NTR}$_3B7C or TetraGFP$^{NTR}$. Live images were taken without fixation or washing steps, i.e., they show directly an NPC-binding in relation to the free concentration and non-specific aggregation with nuclear or cytoplasmic structures. While 3B7C still showed some weak binding to nuclear structures other than NPCs, TetraGFP$^{NTR}$ stains NPCs more crisply, illustrating that TetraGFP$^{NTR}$ is a more specific NPC binder. This experiment was performed twice on independent samples with similar results, and representative images are shown. **c** Plot profiles corresponding to lines across nuclei marked in **b**. Fluorescence (GFP) intensities are normalized to the maximum (arbitrarily set to 100) of each line.

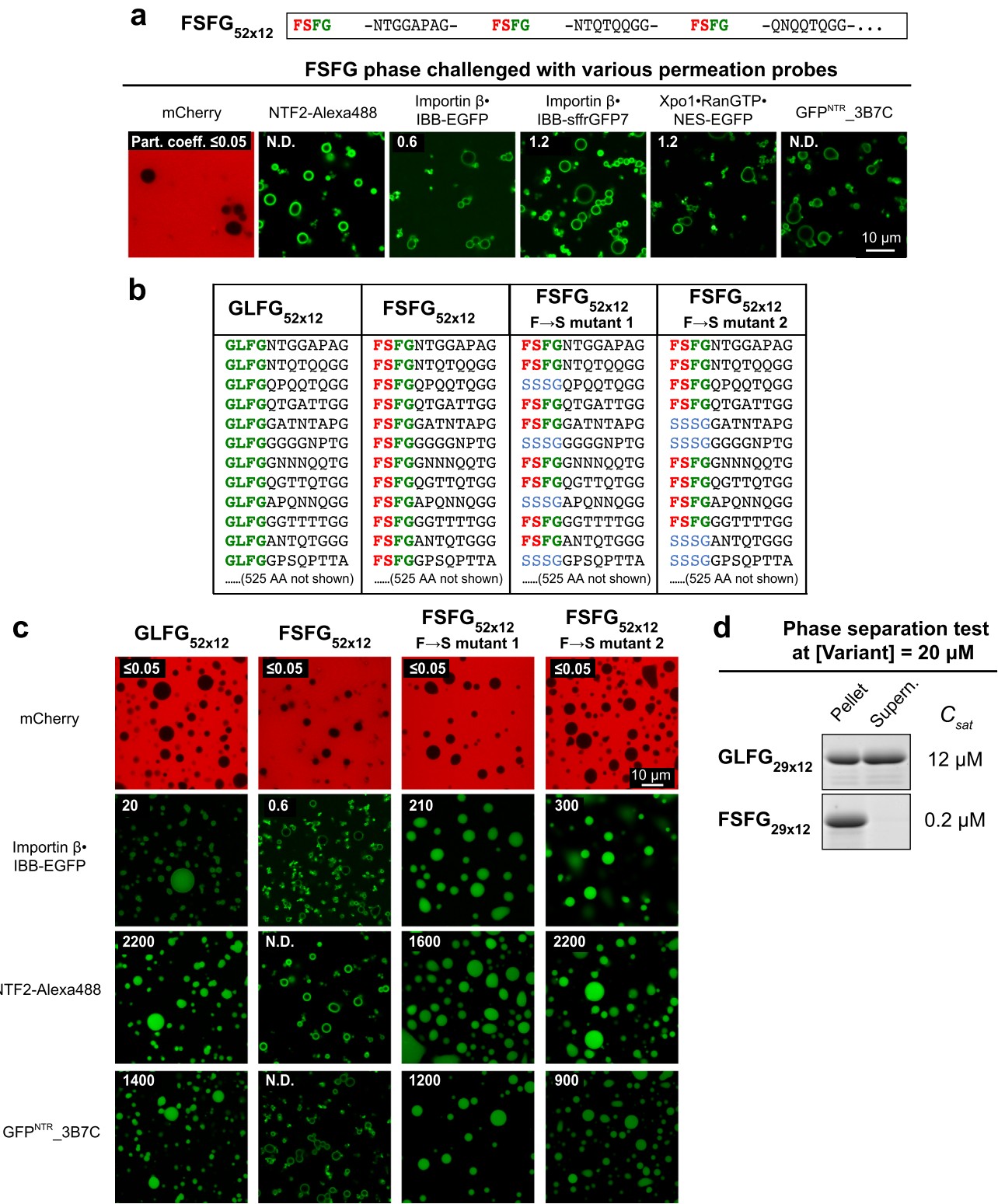

**Fig. 9 | Exchanging GLFG to FSFG motifs results in a hypercohesive FG phase that precludes entry of NTR-cargo complexes. a** Based on a framework of GLFG$_{52×12}$, FSFG$_{52×12}$ was constructed as described above. The FG phase assembled from FSFG$_{52×12}$ was challenged with the indicated probes. Scanning settings/ image brightness were adjusted individually due to the large range of signals. N.D.: Not determined due to difficulties in defining the central region. Note that all probes only arrested at the periphery of the phase. **b** Based on FSFG$_{52×12}$, two mutants with a reduced number (=35 or 36) of FSFG motifs were constructed. "Mutant 1" corresponds to a sequence where every third FSFG motif in FSFG$_{52×12}$ was mutated to SSSG (FG-phobic). "Mutant 2" is similar, but every fifth and sixth FSFG motif were mutated. For space economy, only the N-terminal -144 residues of each are shown. The C-terminal sequences follow the same design strategy as the region shown (see Supplementary Note 1 for complete sequences). Corresponding regions of GLFG$_{52×12}$ and FSFG$_{52×12}$ are shown for comparison. **c** FG phases assembled from the above variants were challenged with the indicated probes as in **a**. **d** GLFG$_{29×12}$ and FSFG$_{29×12}$: corresponding to the N-terminal 29 repeats of GLFG$_{52×12}$ and FSFG$_{52×12}$, respectively, were tested for phase separation at a concentration of 20 μM. Each of the assays was performed twice on independent samples with similar results, and representative images are shown. Note that the FSFG motifs lead to stronger phase separation propensity than the GLFG motifs (see also Supplementary Fig. 2).

**a**

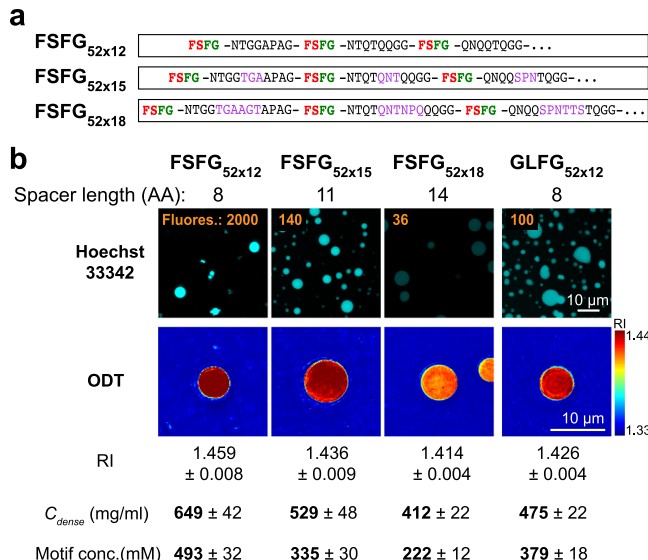

**b**

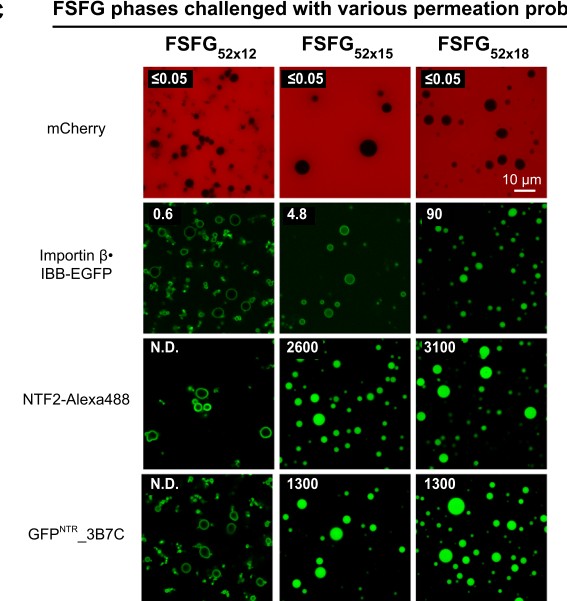

**c**

**Fig. 10 | Hypercohesion of FSFG motifs can be balanced by longer inter-FG spacers. a** Based on FSFG$_{52\times12}$, two variants with the same number (52) of FSFG motifs but longer inter-FSFG spacers were constructed (FSFG$_{52\times15}$ and FSFG$_{52\times18}$). **b** Upper panels: condensed phases assembled from the indicated variants were stained with Hoechst 33342. The numbers in orange indicate the fluorescence intensities of the Hoechst dye inside the FG phases. The fluorescence intensities are relative to that of GLFG$_{52\times12}$ (arbitrarily set to 100). The above assay was performed twice on independent samples with similar results, and representative images are shown. Lower panels: the condensed phases were analysed by optical diffraction tomography (ODT) independently in the absence of Hoechst. Panels show maps of refractive index (RI). For each, ten independent FG particles were analysed. Representative images and mean values (±S.D. between the ten FG particles) of RI, mass density ($C_{dense}$), and FG motif concentration are shown. **c** Condensed phases assembled from FSFG$_{52\times12}$ and its two variants with longer spacers were challenged with the indicated probes. Note that lengthening the spacers relaxes the permeation selectivity.

The preferred overall length of Nup98 FG domains probably relates to providing enough FG mass per NPC and a sufficient contour length for spanning the central NPC channel. Likewise, the phase separation propensity depends on the number of repeats[62].

We now found that the absence of negative charges is a fundamental requirement for inter-repeat cohesion. Similarly, the hydrophobicity and FG motif density are crucial. Replacing the GLFG with more hydrophobic FSFG motifs or increasing the FG motif density by shrinking the inter-FG spacers resulted in an intriguing phenotype, namely hypercohesive FG phases. This increase in cohesion was evident by orthogonal readouts: (1) a higher phase separation propensity/lowered saturation concentration, (2) an increased protein concentration in the resulting dense phase, and (3) an increased fluorescent signal by Hoechst 33342, which scales not just with the local FG motif concentration but apparently also with the occupancy of cohesive contacts.

The most striking functional consequence of hypercohesion is a stricter barrier−to the extent that NTR·cargo complexes or even cargo-free NTRs get arrested at the FG phase's surface and fail to enter. An interpretation is given by the selective phase model[14], which assumes that FG-mediated inter-repeat contacts are obstacles blocking the undesired passage of inert materials. NTRs bind FG motifs, thereby resolving inter-repeat contacts in their immediate vicinity and thus catalysing their own passage. In a hypercohesive phase, these contacts are stabilised by a higher energetic penalty than the FG-binding of NTRs can release. As a result, even NTRs remain excluded. If this happened in living cells, then transport of cargoes would get blocked−with detrimental consequences. This consideration explains the evolutionary pressure for keeping the parameters 'FG motif density' and 'hydrophobicity' of the Nup98 FG domains within their remarkably narrow tolerance.

### Arginines appear to promote NPC passage also by general hydrophobic and H-bonding interactions

Lysine and arginine have long hydrophobic sidechains with positive terminal charges; both can engage in salt bridges and are highly soluble in water. Yet, the two have opposite effects on NPC passage: Exposed lysines render a mobile species slow in NPC passage, while arginines can drastically increase the rate of NPC passage[45]. This behaviour is well recapitulated with in vitro assembled FG phases (Fig. 7), where, e.g., the sffrGFP4 variant (with 25 exposed Arg) shows a 100-fold higher partition coefficient than its 25-lysine counterpart.

This was so far best explained by the positively charged guanidinium group of arginine engaging in cation-π interactions[78] with the phenylalanines of the FG domain. Lysine can engage in similar interactions; however, the planar guanidinium of arginine is favoured here because of better shape complementarity to the planar phenylalanine ring. Furthermore, a planar guanidinium is more easily released from the tetragonal hydrogen-bonding network of water than the (tetragonal) ammonium group of lysine.

We have now characterized an analogous dense phase, the GLLG//L phase, whose phase separation is driven by the hydrophobicity of leucines. It accumulated sffrGFP4$^{NTR}$ 80-fold higher than the 25 R→K variant (Fig. 7). This is a similar selectivity factor as found for the analogous FG or YG phase. However, since this domain contains neither phenylalanines nor tyrosines, we can exclude cation-π interactions as an explanation.

We suspect two kinds of alternative interactions. First, arginines might engage more easily also in general hydrophobic interactions (i.e., between the aliphatic portions of its side chain and the three leucines and other aliphatic groups in the repeats)−again because its planar guanidium group interacts less favourably with water than the lysyl ammonium. Second, the guanidinium group is an excellent hydrogen bond donor and probably bonds to carbonyl oxygens from the polypeptide backbone as well as from asparagine or glutamine sidechains of the phase. The extremely high concentration of repeat units of nearly 400 mg/ml would favour such interactions. Taken together, this suggests a mixed-mode of interaction of arginines with

the in vitro assembled FG phases and—by extrapolation— also during passage through the permeability barrier of NPCs.

Although extremely depleted in FG domains, arginines are common in other phase-separating intrinsically disordered domains, where (cation-π) interactions with tyrosines/ phenylalanines have been reported[64,73,79]. Based on the above considerations, it might be safe to assume that also in these cases, arginines engage in general hydrophobic and H-bonding interactions as well.

Extrapolating further, it appears likely that FG phases are condensed by a variety of simultaneous interactions that include hydrophobic contacts of phenylalanines[27,68], of aliphatic carbons[33] as well as hydrogen-bonding interactions of various sidechain and backbone moieties—even when cross-β-structures[41,69] are not detectable[33].

## Why is the NPC permeability barrier formed by FG and not by YG repeats?

We previously reported that F→Y mutated Nsp1 FG repeats can complement an otherwise lethal FG domain deletion[27]. This is best explained by this YG mutant being able to contribute to the NPC permeability barrier, at least in the presence of the remaining eight FG domains. We now found that an FG→YG mutated domain not only phase-separates very well, but also, the resulting dense phase perfectly excludes mCherry while strongly accumulating the four times larger GFP$^{NTR}$_3B7C variant (Figs. 6–8). This remarkably high transport selectivity poses the question of why the NPC barrier relies on FG but not on YG repeats.

Resilins[80], a class of naturally occurring YG repeat proteins, might provide one answer. These extracellular elastomers store mechanical energy to allow dragonflies a very energy-efficient flight and fleas their remarkable jumping abilities. Resilins mature by a radical-induced formation of covalent Y-Y bonds between the YG repeat units. The same reaction within the NPC permeability barrier (e.g., under conditions of oxidative stress) would lead to an irreversible clogging of the barrier. Since NPC are very long-lived structures (at least in terminally differentiated cells), even a slow rate of this reaction would deteriorate NPC function over time. Thus, the inherent propensity of the phenolic tyrosine side chains to covalent crosslinking could explain the strong selection against tyrosines in Nup FG repeat domains.

Other reasons might be that tyrosines are more promiscuous than phenylalanines in their interactions[45], that YG-rich proteins, such as the yeast prion Sup35[81,82], are prone to form potentially detrimental amyloid-like structures, and that an early co-evolutionary adaptation of NTRs to FG repeats was a hurdle for a later switch to YG repeats. A final reason is to avoid a mistargeting to peroxisomes, which have recently been suggested to rely on a YG phase for controlling their import of proteins[83].

## FG phase arrangement in NPCs

Phase separation of a Nup98 FG domain in bulk buffer leads to a dense phase with spherical FG particles (Figs. 2–10), suggesting that surface tension minimizes the contact area between the hydrophobic FG phase and the surrounding water. These FG particles also feature a rather homogenous protein concentration, indicating that the phase has reached thermodynamic equilibrium. We now measured this protein concentration by ODT and found it to be remarkably high—in the 450 mg/ ml range for a repeat domain with standard FG density (GLFG$_{52×12}$ or wild-type MacNup98A FG domain). This number has interesting implications for the organization of the permeability barrier in intact NPCs, which—according to the latest structures[8,9]—has to span a central channel with a diameter of 60 nm.

Nup98 occurs in 48 copies per NPC, has a ~50 kDa FG domain, and contributes thus 2.5 Megadalton FG mass. If the Nup98 FG phase would contract to its preferred concentration of ~400 mg/ml, then this mass would allow only a 3.5 nm thick disc to span the NPC's cross-section. Such a thin layer would allow for a very rapid NTR passage.

However, it would be rather unstable because surface tension would force such a thin disc into a shape with a smaller surface area, ultimately leading to barrier collapse.

The Nup98 FG phase volume probably increases by Gle2/Rae1 binding the GLEBS domain, the presence of the Nup98 anchor domain, and partitioning NTRs that all occupy volume. Nevertheless, we assume additional stabilization mechanisms to be in place. First, single-molecule tracking has identified a sizeable obstacle in the very centre of the NPC channel, which NTR-cargo complexes circumnavigate[19]. This central object might function in 'spanning' a thin Nup98 FG phase layer across the channel (like the hub in a spoked wheel). This is speculation at the moment, but it can explain the presence of such an object and how a thin barrier layer could be stabilized. A further stabilization might arise from GLFG repeats binding to the NPC scaffold[84].

Finally, ten other FG Nups (Nups 358, 214, 153, 62, 58, 54, 50, 42, Pom121, and Ganp in humans) provide another 12 Megadalton in FG mass. This increases the FG phase volume and should reduce the shape-changing effect of surface tension. We do not expect a homogenous mixing between all these FG domains because their anchor points already dictate a spatial separation and because these domains come with a range of cohesivity. The most cohesive ones will co-phase-separate, forming the barrier's core. The less cohesive and more water-soluble ones, however, should stay more peripheral, reduce (in a surfactant-analogous manner) the surface tension at the water-FG interface, and thereby stabilize the arrangement of a thin cohesive FG layer further. A nice illustration of this principle is soap allowing water to form thin films on flat plastic rings or as soap bubbles.

## Methods

### Nomenclature

All sequence-regularized variants of the Nup98 FG domain in this study are named XXXX$_{NxM}$, where XXXX is the sequence of a perfectly repeated tetrapeptide motif (e.g., GLFG or FSFG), $N$ is the number of the repeat unit, and $M$ is the number of amino acid residues per repeat unit (i.e., $M$ = inter-tetrapeptide spacer length ($L$) + 4). Full sequences of variants are shown in Supplementary Note 1.

### DNA sequences of FG domain variants

DNA sequences encoding the variants were codon-optimised for *E. coli* expression. DNA fragments were synthesized by GenScript and cloned into a bacterial expression vector for overexpression and purification (see below). The *Tetrahymena thermophila* GLEBS domain (44 amino acid residues) of the Mac98A FG domain was included in the amino acid sequences (inserted between the 29th and 30th repeat unit) of most of the variants because the presence of the corresponding non-repetitive DNA sequence improved the ease of DNA syntheses.

### Recombinant protein expression and purification

*Mac98A FG protein domain and variants:* The protein domains were recombinantly expressed in histidine-tagged form with an additional C-terminal cysteine each and purified as described[46]. The proteins were expressed in *E. coli* NEB Express at 30 °C for 4 h with induction by 0.4 mM IPTG. The Mac98A FG domain and most of the FG domain variants (except those specified below) phase-separated in vivo to form inclusion bodies. Cells were resuspended in cold 50 mM Tris/HCl pH 7.5, 300 mM NaCl, 1 mg/ml lysozyme, and lysed by a freeze-thaw cycle followed by mild sonication. Inclusion bodies were recovered by centrifugation (k-factor: 3158; 10 min) and washed once in 50 mM Tris/ HCl pH 7.5, 300 mM NaCl, 5 mM DTT. The FG domain/variant was extracted with 40% formamide, 50 mM Tris/HCl pH 7.5, 10 mM DTT. The extract was cleared by ultracentrifugation (k-factor: 135; 90 min) and applied for 3 h at room temperature to a Ni(II) chelate column. The column was washed in extraction buffer, followed by 200 mM ammonium acetate pH 7.5 (as a volatile buffer). The target protein was

eluted with 30% acetonitrile, 265 mM formic acid, 10 mM ammonium formate, and lyophilized directly. The lyophilized material was weighted for quantification. Unless specified otherwise, the lyophilized material was dissolved to 1 mM protein concentration in 4 M guanidinium hydrochloride (GuHCl).

GLFG//D$_{52×12}$ and GLLG$_{52×12}$: Each was recombinantly expressed with an N-terminal His$_{14}$-ZZ-scSUMO tag (which improved the expression) in *E. coli* NEB Express at 30 °C for 4 h with induction by 0.4 mM IPTG. Cells were resuspended and lysed as described above. However, these target proteins remained in the soluble fraction of cell lysate and did not form inclusion bodies in vivo, indicating a lack of cohesiveness. The soluble fraction of cell lysate was cleared by ultracentrifugation and applied directly to a Ni(II) chelate column. The column was washed extensively in 50 mM Tris/HCl pH 7.5, 300 mM NaCl, 20 mM imidazole, 20 mM DTT, and then in protease buffer: 50 mM Tris/HCl pH 7.5, 300 mM NaCl, 5 mM DTT. 50 nM SUMO protease in protease buffer was applied for overnight on-column cleavage[85,86]. The cleaved targets eluted with the protease, remained soluble, were re-buffered to 30% acetonitrile on a PD10 Sephadex column (GE Healthcare), and finally lyophilized.

*NTRs, EGFP, GFP variants, and mCherry:* Most were expressed as His-tagged-fusions (Supplementary Table 3) and purified by native Ni(II) chelate chromatography, as described previously[32,45]. Elution was performed by on-column protease cleavage[85,86].

## Analysis of phase separation by centrifugation

For each, 1 µl of a fresh stock of FG domain or variant (typically 1 mM protein in 4 M GuHCl; for GLFG$_{29×12}$ and FSFG$_{29×12}$: 2 mM protein in 4 M GuHCl) was rapidly diluted with assay buffer (50 mM Tris/HCl, 5 mM DTT, 150 mM NaCl, unless specified otherwise), to the concentration(s) stated in the figures, at 25 °C. After incubation for 1 min, the FG phase (insoluble content) was pelleted by centrifugation (21,130 × *g*, 30 min, using a temperature-controlled Eppendorf 5424 R centrifuge equipped with a FA-45-24-11 rotor) at 25 °C. Equivalent ratios of the pellet (condensed FG phase) and the supernatant were analysed by SDS-PAGE/Coomassie blue-staining (the exact amount loaded for SDS-PAGE was adjusted individually such that the loaded amount of pellet + supernatant equalled 4.1 µg). Saturation concentration (a.k.a. threshold concentration or critical concentration for phase separation) of a given sample was taken as the concentration that remained in the supernatant, which was estimated with a concentration series loaded onto the same gel. All tests were performed at least two times independently with similar results. Representative gel images are shown.

## Optical diffraction tomography (ODT)

For each, 1 µl of a fresh stock of FG domain or variant (typically 1 mM protein in 4 M GuHCl; for GLFG$_{52×15}$ 1 mM protein in 2 M GuHCl), containing 0.5% molecules coupled with Atto488 (ATTO-TEC, Germany) via C-terminal cysteine, was rapidly diluted with assay buffer (500-fold for GLFG$_{52×13}$; 300-fold for GLFG$_{52×14}$ and GLLG//L$_{52×12}$; 100-fold for GLFG$_{52×15}$ and 600-fold for all others). See ref. [33] for the preparation of Atto488-coupled FG domains. The resulting mixture was placed on µ-slide 18-well (IBIDI, Germany), which had been passivated with 0.1 mg/ml Maltose-binding protein (MBP). FG particles were allowed to sediment under gravity for 15 mins and analysed with a custom-built correlative epifluorescence-ODT microscopy setup[61]. FG particles emitting Atto488-fluorescence in the well were visualised and located by epifluorescence microscopy coupled with the setup. The three-dimensional (3D) refractive index (RI) was measured using optical diffraction tomography (ODT) employing Mach-Zehnder interferometry similar to the one described[60,61,87,88] by a coherent laser beam with a wavelength of 532 nm. This interferometric technique measured spatially modulated holograms from 150 different angles, from which the complex optical fields were retrieved. By mapping the Fourier spectra of retrieved complex optical fields onto

the surface of the Ewald sphere in the 3D Fourier space according to the Fourier diffraction theorem, 3D RI tomograms were reconstructed. Detailed principles for tomogram reconstruction can be found in refs. [59,89,90]. The image acquisition, field retrieval, and RI tomogram reconstruction were performed using custom-written MATLAB scripts (R2020a). Several reconstruction algorithms have also been developed for reconstructing RI tomograms of samples encountering multiple light scattering[91,92], which can extend the characterization of samples with heterogeneous density distribution or higher mass density if required. The mean RI value of each sample was measured by manually segmenting regions of interest (ROIs) from the central slice of the reconstructed tomogram using FIJI 2.9.0[93]. To calculate the mass density of each ROI, we used the following relationship[87]:

$$n = n_{medium} + \alpha C \qquad (2)$$

where $n$ = measured RI in the ROI, $n_{medium}$ = RI of the assay medium (1.336, measured with visible light by an Abbe refractometer: ABBE-2WAJ from Arcarda), $\alpha$ = RI increment (0.190 ml/g for proteins[94]) and $C$ = mass density in the ROI.

## Calculation of solvent contents

$$\text{Solvent content (\%)} = (1 - C_{dense} \cdot 0.70 \text{ml/g}) \times 100\% \qquad (3)$$

*Data fitting:* Fittings in Fig. 2 were by the least squares method by Microsoft Excel 16.42. Mean values were fitted.

## FG phase preparation for permeation assays

For each, 1 µl of a fresh stock of FG domain or variant (1 mM unlabelled protein in 4 M GuHCl, unless specified elsewhere) was rapidly diluted with 50 µl assay buffer, and 7.5 µl of the suspension was mixed with 22.5 µl substrate. As described[46], the substrate contains either 12 µM Hoechst 33342 (Thermo Scientific)/6 µM mCherry/3 µM EGFP/3 µM efGFP_8Q/3 µM efGFP_8R/1 µM of an NTR or [1.5 µM NTR pre-incubated with 1 µM cargo], in assay buffer. Note: before mixing NTF2 with RanGDP-Atto488, the Ran protein was incubated with 2 µM RanGAP for 30 mins, which converts all Ran to the GDP-bound form in the assay. The resulting mixture was placed on collagen-coated µ-slides 18-well (IBIDI, Germany). FG particles were allowed to sediment under gravity for 1 h before imaging by CLSM. The samples for ODT and CLSM were prepared independently.

## Confocal laser scanning microscopy (CLSM) and quantification of partition coefficients

Hoechst 33342, mCherry, and GFP/Alexa488 signals were acquired (as described[45,46]) with 405, 561, and 488 nm excitation, respectively, with a Leica SP5 confocal scanning microscope equipped with a 63× oil immersion objective and hybrid detectors (standard mode, in which non-linear response of the detector was auto-corrected) at 21 °C. Scanning settings (e.g., laser power) were adjusted individually such that the quantification was within the dynamic range. Typically, the powers of the 405, 561, and 488 nm lasers were set to 1–5%, 20%, and 2–8%, respectively. For optimal signal-to-noise ratios, line-averaging was applied in all cases. Frame accumulation was additionally applied for recording weak signals, e.g., when the probe was excluded by the FG phase.

To determine the Hoechst fluorescence signals, raw signals in the centres of 3–5 (in focus) FG particles were quantified using Leica Application Suite X 3.3.0. Standard deviations were typically about 10-15% between individual particles. For validation, the mean signal of a given sample was compared to that of GLFG$_{52×12}$, which was recorded under the same settings. To determine the partition coefficients, raw signals in the centres of 3–5 (in focus) FG particles (IN) and three reference areas within the aqueous region (OUT) were quantified using the same software. If necessary, XZ scans were acquired to determine

the signals along the equators of FG particles. Partition coefficients (Part. Coeff.) for each particle were calculated according to: Part. Coeff. = IN / Mean (OUT). As reported previously[45], standard deviations were typically <10% between individual particles and between experiments. Mean values are shown in the figures. All measurements were performed at least two times independently with similar results. Representative images are shown in the figures.

### NPC-staining tests of GFP[NTR] variants
These were performed two times independently as described[45]—with similar results. Figure 8b shows representative images.

### Reporting summary
Further information on research design is available in the Nature Portfolio Reporting Summary linked to this article.

## Data availability
The data generated in this study are provided in the Supplementary Information/Source Data file. Source data are provided with this paper.

## Code availability
Custom MATLAB scripts for ODT reconstruction are publicly available at: https://github.com/OpticalDiffractionTomography.

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

## Acknowledgements

The authors wish to thank Waltraud Taxer and Susanne Brandfass for technical help, Tom A. Rapoport (Harvard Medical School) for critical reading, Kyoohyun Kim and Jochen Guck (Max Planck Institute for the Science of Light) for their support of the ODT setup, as well as the Max-Planck-Gesellschaft, the Deutsche Forschungsgemeinschaft (SFB 860 and SFB 1190 to D.G.), and the Joachim-Herz-Stiftung (Add-On Fellowship to A.B.) for funding.

## Author contributions

S.C.N. planned and conducted most of the experiments. A.B. conducted ODT measurements and data analysis. T.H. designed the TetraGFP[NTR] variant. J.S. conducted preliminary experiments with YG phases. D.G. conceived the overall concepts of the study and wrote the manuscript. S.C.N., S.R., and D.G. contributed to experimental design, data analysis, and interpretation.

## Funding

## Competing interests

The authors declare no competing interests.
