## [Peer review file · Nature Communications]

REVIEWER COMMENTS

Reviewer #1 (Remarks to the Author):

Transport across the nuclear envelope is mediated by nuclear pore complexes, large toroidal structures with a central channel. The channel is filled with FG-repeat domains, intrinsically disordered protein regions anchored to the NPC scaffold. We know the principal character of the hydrogel and how it confers selectivity of the NPC, however, we do not know it in great detail. In its longstanding effort to elucidate the mechanistic details of the FG-hydrogel, here, the Görlich lab provides yet another stunning advance. The authors generate an FG-hydrogel from 52 concatenated copies of a GLFG repeat within a 12 aa sequence, which behaves essentially equivalently to its parental Nup98A FG domain from tetrahymena, in itself a mimic of the natural FG-barrier in the NPC. In a systematic analysis, the authors dissect the importance of the FG-repeat motif, the spacer length, and a number of other parameters to drill down on how nuclear transport is regulated on a very fundamental level. Many important aspects of the FG-hydrogel and its interaction with NTRs and other proteins are revealed here, and these will undoubtedly shape the way forward to understanding nucleocytoplasmic transport in much better detail. This study is also of fundamental interest to the wider field of phase separation, as the FG-hydrogel was not only the first to be described, but is probably also the best understood.

The experiments are carried out at a very high standard and the manuscript is written clearly and succinctly.

We highly recommend publication without delay and only have minor suggestions for the authors to consider.

1. It would be interesting if the authors could go into some depth about what these results may mean for the FG-hydrogel in vivo. For example, the volume of the central channel in the NPC will not have a ~200mg/ml concentration of FG domain throughout, therefore presumably only parts contain hydrogels? Are they separated between different FG-Nups? What might it mean that the hydrogel is not uniform? Do the recent NPC structures help in deciphering possible heterogeneity within the natural FG-hydrogel? These issues were discussed already in Schmidt and Görlich, eLife 2015, but we wonder how much more can be said now, with the results of this paper, and the knowledge that the central channel is ~50% wider than previously estimated.

2. The values for C(dense), FG motif density, and solvent % in Figure 2 should be reported with measured errors, or at least reasonable estimates.

3. In Figure 1d, the mutations could be highlighted better. We think it would be more intuitive if the GLFG boxes would actually be moved closer or further apart, for shorter and longer spacers, respectively. Nothing substantive, just a better visual.

4. Also in Figure 4, the mutations should stand out a bit better. The bright yellow highlight of the repeats takes away from the variations themselves.

Line 78. Eliminate the 'A'

Line 89-90. 'GFP has been evolved' better 'GFP variants have been developed'

Line 305. GiFG should read GIFG

In Figure 4: GiFG label should read GIFG

In Figures 5-8 we think 'FG domain variants' would fit better than 'FG repeat variants' for the title, as it is the domains, rather than the repeats that form the droplets.

Reviewer #2 (Remarks to the Author):

Ng et al. present a systematic investigation of the Nup98 sequence features, crucial to the NPC permeability barrier and report that shortening inter-FG spacers enhances cohesion, increases phase density, and tightens such barrier. The authors have performed highly comprehensive experiments and analyses, exploiting various experimental techniques, including confocal laser scanning microscopy (CLSM) and optical diffraction tomography (ODT). The topic of the manuscript is very important, the approaches are well designed, and the results are interesting. In principle, I am in support of the publication of the manuscript. However, there are a few important comments to be addressed.

- Figure 1. It seems that the images in Fig. 1a and 1b are from different samples. I wonder if the authors have measured both the CLSM and ODT results for the same samples.

- As far as I know, the assumption of ODT is based on weak scattering. When the refractive index of (RI) contrast between a sample and a surrounding medium is smaller than ~ 0.1 , it would be OK. However, suppose the difference is greater than 0.1, due to multiple scattering of light inside a sample. In that

case, the assumption of weak scattering fails, and results in the underestimation of RI values and/or the generation of artifacts in tomographic reconstruction. For example, when the spacer length is longer than 8, I think the measured values would be valid. But, when the spacer length is shorter, like 6 or 7, the RI difference is greater than 0.1, and I would expect the assumption of ODT reconstruction may not be valid. Recently, to consider multiple light scattering effects in ODT, several groups develop the reconstruction algorithms, e.g. "Learning approach to optical tomography." *Optica* 2.6 (2015): 517-522 or Inverse problem solver for multiple light scattering using modified Born series." *Optica* 9.2 (2022): 177-182. I wonder whether the authors take into account the effects of multiple light scattering?

- The sentence in Lines 320-322 may need further explanation.

- Line 488, please cite the general review paper on the relevant field: "Quantitative phase imaging in biomedicine." *Nature photonics* 12.10 (2018): 578-589.

- Line 497, Please cite the related technique: Barer, R. "Determination of dry mass, thickness, solid and water concentration in living cells." *Nature* 172.4389 (1953): 1097-1098.

- Line 498: What is the wavelength used for the Abbe refractometer. Is the wavelength of the refractometer the same as the used ODT system?

- Lines 519-520. Can the authors provide more detailed information about the ranges of laser power and measurement time?

- Please define the abbreviations at their first uses.

Reviewer #3 (Remarks to the Author):

The authors have created an idealized FG-Nup system (previously reported in a couple of papers) that allows them to systematically interrogate the grammar of the FG-Nup interactions and how these code for the selection/filtration function of their resulting condensates. The authors make multiple interesting findings related to how the selectivity of the NPC is encoded within the repetitive protein sequence. They find that increasing the interaction strength between the repeats increases the

selectivity of the condensates while increasing spacer length increases how permissive the condensate is, providing a physical mechanism underlying selectivity. The hydrophobicity of the repeats was necessary for maintaining phase separation. NTR entry was sensitive to some of these mutations, indicating co-evolution between cohesion of the NPC and selectivity for NTRs. Using orthogonal probes derived from GFP, they interrogated the surface properties compatible with entry and found that arginines promoted entry into the condensates through interactions other than cation- π . The author's results support their model that the interplay between the cohesion and mesh structure of the condensates, determines the condensate's selectivity. The surface characteristics of the NTR mediate interaction with the network and can locally dissolve it, promoting the entry of its cargo.

This manuscript beautifully, systematically and quantitatively disentangles the molecular mechanism of how Nup98-derived condensates can mediate transport selectivity similar to the NPC. It will be of high interest to the NPC as well as the phase separation community, and I support publication after the following comments are addressed.

Specific comments:

1. The authors generate FG repeat particles via phase separation and use their ability to exclude or enrich other proteins to understand the molecular determinants of the semipermeable barrier of the NPC. This seems to work well overall. However, a clear conceptual difference is that in the NPC, the FG Nups are anchored in the scaffold and the concentration of FG repeats in the pore is determined by the stoichiometry in the pore and pore dilation rather than by the dense phase concentration resulting from phase separation. The authors should address these conceptual differences and which limitations their conclusions thus have (if any).
2. The authors should reconsider their usage of the term "evolved" for their designed GFP constructs. In their previous paper (Citation #33), they described these constructs as engineered.
3. I find "phases" to be a problematic term for the condensates formed by the FG repeat proteins. Phases presumably refers to at least two phases, and in this manuscript, I would most naturally understand them as being the dilute and the dense phase.
4. Please add citations when discussing the chemical properties of arginine vs lysine. (Paragraphs 2 and 3 of the section labeled "Arginines promote FG phase entry not just through cation- π interactions")
5. In the 2nd paragraph of "Phase separation tolerates FG motif mutation if overall hydrophobicity is preserved" the authors state that they "tested several additional motifs with similar hydrophobicity" and "observed similar saturation concentrations". However, the range of saturation concentrations observed spans an order of magnitude (<0.04 to 0.5 micromolar) This is dissimilar enough to not warrant some explanation. This should be explainable by referencing existing literature regarding tyrosine's stronger interaction strength and the increased solubility of the substitutions, particularly given the sequences are repeated 52 times.
6. A few concluding paragraphs knitting the conclusions of the manuscript together and placing them in the larger context of the literature would be helpful. This would help readers understand how the

different aspects of FG-Nup sequences work together to determine selectivity towards NTRs and exclusion of other large molecules.

Minor comments:

1. There are a few instances where the authors use lowercase "i" instead of uppercase "I" for isoleucine, for example in Figure 4.

Major changes to the manuscript:

- We implemented all the small changes suggested by the reviewers.
- We expanded the ODT dataset to the FG-motif variations (Figures 4 & 10).
- We added a brief discussion section (as suggested by Reviewer 3).

Answers to the reviewers' comments (for clarity, we repeat their points in blue before our replies)

Reviewer #1 (Remarks to the Author):

Transport across the nuclear envelope is mediated by nuclear pore complexes, large toroidal structures with a central channel. The channel is filled with FG-repeat domains, intrinsically disordered protein regions anchored to the NPC scaffold. We know the principal character of the hydrogel and how it confers selectivity of the NPC, however, we do not know it in great detail. In its longstanding effort to elucidate the mechanistic details of the FG-hydrogel, here, the Görlich lab provides yet another stunning advance. The authors generate an FG-hydrogel from 52 concatenated copies of a GLFG repeat within a 12 aa sequence, which behaves essentially equivalently to its parental Nup98A FG domain from tetrahymena, in itself a mimic of the natural FG-barrier in the NPC. In a systematic analysis, the authors dissect the importance of the FG-repeat motif, the spacer length, and a number of other parameters to drill down on how nuclear transport is regulated on a very fundamental level.

Many important aspects of the FG-hydrogel and its interaction with NTRs and other proteins are revealed here, and these will undoubtedly shape the way forward to understanding nucleo-cytoplasmic transport in much better detail. This study is also of fundamental interest to the wider field of phase separation, as the FG-hydrogel was not only the first to be described, but is probably also the best understood.

The experiments are carried out at a very high standard and the manuscript is written clearly and succinctly.

We highly recommend publication without delay and only have minor suggestions for the authors to consider.

Thank you very much for this very positive evaluation!

1. It would be interesting if the authors could go into some depth about what these results may mean for the FG-hydrogel in vivo. For example, the volume of the central channel in the NPC will not have a ~200mg/ml concentration of FG domain throughout, therefore presumably only parts contain hydrogels? Are they separated between different FG-Nups? What might it mean that the hydrogel is not uniform? Do the recent NPC structures help in deciphering possible heterogeneity within the natural FG-hydrogel? These issues were discussed already in Schmidt and Görlich, eLife 2015, but we wonder how much more can be said now, with the results of this paper, and the knowledge that the central channel is ~50% wider than previously estimated.

These numbers and the stoichiometries of Nups pose quite interesting constraints, also in the light of the higher protein concentration in the dense Nup98 FG phase. We do agree that this deserves discussion and added the following:

FG phase arrangement in NPCs

Phase separation of a Nup98 FG domain in bulk buffer leads to a dense phase with spherical FG particles (Figs. 2-10), suggesting that surface tension minimizes the contact area between the hydrophobic FG phase and the surrounding water. These FG particles also feature a rather homogenous protein concentration indicating that the phase has reached thermodynamic equilibrium. We now measured this protein concentration by ODT and found it to be remarkably high – in the 450 mg/ml range for a repeat domain with standard FG density (GLFG_{52x12} or wild-type MacNup98A FG). This number has interesting implications for the organization of the permeability barrier in intact NPCs, which – according to the latest structures^{8,9} – has to span a central channel with a diameter of 60 nm.

Nup98 occurs in 48 copies per NPC, has a 50 kDa FG domain, and contributes thus 2.5 Megadalton FG mass. If the Nup98 FG phase would contract to its preferred concentration of 400 mg/ml, then this mass would allow only a 3.5 nm thick disc to span the NPC's cross-section. Such a thin layer would allow for a very rapid NTR passage. However, it would be rather unstable because surface tension would force such a thin disc into a shape with a smaller surface area, ultimately leading to barrier collapse.

The Nup98 FG phase volume probably increases by Gle2/Rae1 binding the GLEBS domain, the presence of the Nup98 anchor domain, and partitioning NTRs that all occupy volume. Nevertheless, we assume additional stabilization mechanisms to be in place. First, single-molecule tracking has identified a sizeable obstacle in the very center of the NPC channel, which NTR-cargo complexes circumnavigate¹⁹. This central object might function in 'spanning' a thin Nup98 FG phase layer across the channel (like the hub in a spoked wheel). This is pure speculation at the moment, but it can explain the presence of such an object and how a thin barrier layer could be stabilized. A further stabilization might arise from GLFG repeats binding to the NPC scaffold⁷⁹.

Finally, ten other FG Nups (Nups 358, 214, 153, 62, 58, 54, 50, 42, Pom121, and Ganp in humans) provide another 12 Megadalton in FG mass. This increases the FG phase volume and should reduce the shape-changing effect of

surface tension. We do not expect a homogenous mixing between all these FG domains because their anchor points already dictate a spatial separation and because these domains come with a range of cohesivity. The most cohesive ones will co-phase-separate, forming the barrier's core. The less cohesive and more water-soluble ones, however, should stay more peripheral, reduce (in a surfactant-analogous manner) the surface tension at the water-FG interface, and thereby stabilize the arrangement of a thin cohesive FG layer further. A nice illustration of this principle is soap allowing water to form thin films on flat plastic rings or as soap bubbles.

2. The values for C(dense), FG motif density, and solvent % in Figure 2 should be reported with measured errors, or at least reasonable estimates.

Done as suggested for Figure 2 as well as for Figures 4c and 10b, which now also contain ODT data for the remaining repeat variants.

3. In Figure 1d, the mutations could be highlighted better. We think it would be more intuitive if the GLFG boxes would actually moved closer or further apart, for shorter and longer spacers, respectively. Nothing substantive, just a better visual.

Implemented as suggested.

4. Also in Figure 4, the mutations should stand out a bit better. The bright yellow highlight of the repeats takes away from the variations themselves.

We have changed the layout of the sequence panels as suggested.

Line 78. Eliminate the 'A'

Done as suggested.

Line 89-90. 'GFP has been evolved' better 'GFP variants have been developed'

Considering a comment from Reviewer 3 (see below) as well, we now changed the words to 'GFP variants have been engineered'

Line 305. GiFG should read GIFG

Changed as suggested.

In Figure 4: GiFG label should read GIFG

Changed as suggested.

In Figures 5-8 we think 'FG domain variants' would fit better than 'FG repeat variants' for the title, as it is the domains, rather than the repeats that form the droplets.

Changed as suggested.

Reviewer #2 (Remarks to the Author):

Ng et al. present a systematic investigation of the Nup98 sequence features, crucial to the NPC permeability barrier and report that shortening inter-FG spacers enhances cohesion, increases phase density, and tightens such barrier. The authors have performed highly comprehensive experiments and analyses, exploiting various experimental techniques, including confocal laser scanning microscopy (CLSM) and optical diffraction tomography (ODT). The topic of the manuscript is very important, the approaches are well designed, and the results are interesting.

Thank you very much!

In principle, I am in support of the publication of the manuscript. However, there are a few important comments to be addressed.

- Figure 1. It seems that the images in Fig. 1a and 1b are from different samples. I wonder if the authors have measured both the CLSM and ODT results for the same samples.

We assume, the reviewer meant Figure 2. The figure contains images from different setups. The Hoechst-signal was recorded by our standard confocal (CLSM) setup (consistent then with all partitioning data). The ODT signal was recorded by a dedicated system. Though we also have confocal images from that ODT setup, we did not show them in the main text but have included a figure with representative images here. The samples measured on the ODT system were prepared separately. We have added a sentence in the Method section to clarify this.

Figure 1: Representative FG particles with varying spacer lengths. The top panel shows confocal fluorescence images, while the bottom panel shows the refractive index (RI) images of the central slice of the FG particles. To visualise the particles via fluorescence, 0.5% of the protein molecules were labelled with Atto488. All images in this figure were acquired on the same optical setup capable of correlative confocal fluorescence and ODT.

- As far as I know, the assumption of ODT is based on weak scattering. When the refractive index of (RI) contrast between a sample and a surrounding medium is smaller than ~ 0.1 , it would be OK. However, suppose the difference is greater than 0.1, due to multiple scattering of light inside a sample. In that case, the assumption of weak scattering fails, and results in the underestimation of RI values and/or the generation of artifacts in tomographic reconstruction. For example, when the spacer length is longer than 8, I think the measured values would be valid. But, when the spacer length is shorter, like 6 or 7, the RI difference is greater than 0.1, and I would expect the assumption of ODT reconstruction may not be valid. Recently, to consider multiple light scattering effects in ODT, several groups develop the reconstruction algorithms, e.g. "Learning approach to optical tomography." *Optica* 2.6 (2015): 517-522 or Inverse problem solver for multiple light scattering using

modified Born series." *Optica* 9.2 (2022): 177-182. I wonder whether the authors take into account the effects of multiple light scattering?

We thank Reviewer #2 for this genuine concern, which we are happy to discuss. Briefly, we have used the first Rytov approximation to reconstruct our RI tomograms. As the reviewer points out, the first Rytov approximation solves the inversion problem of light diffraction induced by weak scattering

objects. The first Rytov approximation is valid when $n_\delta \gg \left(\nabla\phi \frac{\lambda}{2\pi}\right)^2$, where n_δ is the refractive index variation over the length scale of wavelength, $\nabla\phi$ is the phase gradient, and λ is the wavelength of the illumination beam. In our samples, droplet size is small, which generates a lower phase gradient on the sample boundary. In addition, the phase gradient is small due to the homogeneous density distribution within the sample (see representative images below, showing the maximum intensity projections of the phase-separated droplets):

Figure 2: Maximum intensity projections of FG particles with varying spacer lengths.

Further information on the validity of the first Rytov approximation can be found elsewhere (1, 2). In addition, we performed an experiment where we created an emulsion of mineral oil (M8410, Sigma) in distilled water and imaged oil droplets ($n=10$) with our ODT system. As per the manufacturer's datasheet the RI of the mineral oil is 1.467 (also confirmed independently using an Abbe refractometer) and the RI of water is 1.333. Upon using the same reconstruction algorithm that was used for the FG particles we found the average RI of the oil droplets to be 1.4679 ± 0.0051 (Mean \pm SD, Figure 3). Therefore, despite there being a significantly high difference between the RI of the oil droplet and the surrounding medium ($\Delta RI = 0.137$) we were able to reliably reconstruct tomograms with an error of about 0.06%.

Figure 3. ODT reconstruction algorithm reliably obtains the RI of high RI ($\Delta RI > 0.1$) samples in water. Representative image of a mineral oil droplet in water. Graph showing the RI of oil according to the manufacturer (red circle, from the manufacturer's datasheet) and the RI of oil droplets ($n=10$, black squares) as measured by ODT.

Following the reviewer's suggestion, we now write "*Several reconstruction algorithms have been developed for reconstructing RI tomograms of samples encountering multiple light scattering (4, 5), which can extend the characterization of samples with heterogeneous density distribution or higher mass density if required.*" in the Methods.

(1) Habashy, T.M., Groom, R.W. and Spies, B.R., 1993. Journal of Geophysical Research: Solid Earth, 98(B2), pp.1759-1775. doi.org/10.1029/92JB02324

(2) K. Kim, J. Yoon, S. Shin, S. Lee, S. Yang, and Y. Park, J. Biomed. Photonics Eng. 2, 020201 (2016). DOI: [10.18287/JBPE16.02.020201](https://doi.org/10.18287/JBPE16.02.020201)

(3) Biswas, A., Kim, K., Cojoc, G., Guck, J. and Reber, S., 2021. Developmental Cell, 56(7), pp.967-975. doi.org/10.1016/j.devcel.2021.03.013

(4) Lim, J., Ayoub, A.B., Antoine, E.E. and Psaltis, D., 2019. Light: Science & Applications, 8(1), pp.1-12.. doi.org/10.1038/s41377-019-0195-1

(5) Lee, M., Hugonnet, H. and Park, Y., 2022. Optica, 9(2), pp.177-182. doi.org/10.1364/OPTICA.446511

- The sentence in Lines 320-322 may needs further explanation.

We have smoothed the text.

- Line 488, please cite the general review paper on the relevant field: "Quantitative phase imaging in biomedicine." Nature photonics 12.10 (2018): 578-589.

Cited as suggested.

- Line 497, Please cite the related technique: Barer, R. "Determination of dry mass, thickness, solid and water concentration in living cells." Nature 172.4389 (1953): 1097-1098.

Cited as suggested.

- Line 498: What is the wavelength used for the Abbe refractometer. Is the wavelength of the refractometer is the same as the used ODT system?

Light at the visible spectrum was used for the Abbe refractometer, while the wavelength used in the ODT system was 532 nm. This information is now provided in the Methods section.

- Lines 519-520. Can the authors provide more detailed information about the ranges of laser power and measurement time?

More information is now provided in the Methods section.

- Please define the abbreviations at their first uses.

We double-checked the manuscript and introduced 5 new definitions.

Reviewer #3 (Remarks to the Author):

The authors have created an idealized FG-Nup system (previously reported in a couple of papers) that allows them to systematically interrogate the grammar of the FG-Nup interactions and how these code for the selection/filtration function of their resulting condensates. The authors make multiple interesting findings related to how the selectivity of the NPC is encoded within the repetitive protein sequence. They find that increasing the interaction strength between the repeats increases the selectivity of the condensates while increasing spacer length increases how permissive the condensate is, providing a physical mechanism underlying selectivity. The hydrophobicity of the repeats was necessary for maintaining phase separation. NTR entry was sensitive to some of these mutations, indicating co-evolution between cohesion of the NPC and selectivity for NTRs. Using orthogonal probes derived from GFP, they interrogated the surface properties compatible with entry and found that

arginines promoted entry into the condensates through interactions other than cation- π . The author's results support their model that the interplay between the cohesion and mesh structure of the condensates, determines the condensate's selectivity. The surface characteristics of the NTR mediate interaction with the network and can locally dissolve it, promoting the entry of its cargo.

This manuscript beautifully, systematically and quantitatively disentangles the molecular mechanism of how Nup98-derived condensates can mediate transport selectivity similar to the NPC. It will be of high interest to the NPC as well as the phase separation community, and I support publication after the following comments are addressed.

Thank you very much for this very positive evaluation.

Specific comments:

1. The authors generate FG repeat particles via phase separation and use their ability to exclude or enrich other proteins to understand the molecular determinants of the semipermeable barrier of the NPC. This seems to work well overall. However, a clear conceptual difference is that in the NPC, the FG Nups are anchored in the scaffold and the concentration of FG repeats in the pore is determined by the stoichiometry in the pore and pore dilation rather than by the dense phase concentration resulting from phase separation. The authors should address these conceptual differences and which limitations their conclusions thus have (if any).

We would assume that the FG repeat concentration in the phase is mostly determined by the phase separation properties – an unknown being how much mixing between different FG domains occurs and to what extent partitioning NTRs and other FG ligands change the concentration. The geometry and stoichiometry of anchorage are, of course, crucial constraints. Since this point was also raised by Reviewer 1, we have added accordingly a section to the Discussion (see above, answers to Reviewer 1).

2. The authors should reconsider their usage of the term “evolved” for their designed GFP constructs. In their previous paper (Citation #33), they described these constructs as engineered.

Changed as requested.

3. I find “phases” to be a problematic term for the condensates formed by the FG repeat proteins. Phases presumably refers to at least two phases, and in this manuscript, I would most naturally understand them as being the dilute and the dense phase.

Corrected in several instances.

4. Please add citations when discussing the chemical properties of arginine vs lysine. (Paragraphs 2 and 3 of the section labeled “Arginines promote FG phase entry not just through cation- π interactions)

In the initial submission, we already cited reference 70 (now reference 77; Crowley PB, Golovin A (2005) Cation- π interactions in protein-protein interfaces. *Proteins*, 59: 231–239). We still think that this deep dive into the topic is a very appropriate reference.

5. In the 2nd paragraph of “Phase separation tolerates FG motif mutation if overall hydrophobicity is preserved” the authors state that they “tested several additional motifs with similar hydrophobicity” and “observed similar saturation concentrations”. However, the range of saturation concentrations observed spans an order of magnitude (<0.04 to 0.5 micromolar) This is dissimilar enough to not warrant some explanation. This should be explainable by referencing existing literature regarding tyrosine’s stronger interaction strength and the increased solubility of the substitutions, particularly given the sequences are repeated 52 times.

Indeed, differences in solubility get greatly amplified by having 52 repeats (given that solubility drops exponentially with repeat number). In this light, the differences are rather small. We expanded the discussion on this and included a few citations.

6. A few concluding paragraphs knitting the conclusions of the manuscript together and placing them in the larger context of the literature would be helpful. This would help readers understand how the different aspects of FG-Nup sequences work together to determine selectivity towards NTRs and exclusion of other large molecules.

We have added a Discussion as suggested.

Minor comments:

1. There are a few instances where the authors use lowercase “i” instead of uppercase “I” for isoleucine, for example in Figure 4.

Amended.

REVIEWERS' COMMENTS

Reviewer #1 (Remarks to the Author):

All my concerns have been adequately addressed. I recommend publication without further changes.

Reviewer #2 (Remarks to the Author):

The authors have adequately addressed the raised comment. I strongly support the publication of the manuscript.

Reviewer #3 (Remarks to the Author):

The authors have addressed my comments satisfactorily. This manuscript beautifully, systematically and quantitatively disentangles the molecular mechanism of how Nup98-derived condensates can mediate transport selectivity similar to the NPC. It will be of high interest to the NPC as well as the phase separation community, and I support publication.